# LoRC: Low-Rank Compression for LLMs KV Cache with a Progressive Compression Strategy

## Abstract

The Key-Value (KV) cache is a crucial component in serving transformer-based autoregressive large language models (LLMs), enabling faster inference by storing previously computed KV vectors. However, its memory consumption scales linearly with sequence length and batch size, posing a significant bottleneck in LLM deployment. Existing approaches to mitigate this issue include: (1) efficient attention variants integrated in upcycling stages, which requires extensive parameter tuning thus unsuitable to pre-trained LLMs; (2) KV cache compression at test time, primarily through token eviction policies, which often overlook inter-layer dependencies and can be task-specific.

This paper introduces an orthogonal approach to KV cache compression. We propose a *low-rank approximation* of KV weight matrices, allowing for plug-in integration with existing transformer-based LLMs without model retraining. To effectively compress KV cache at the weight level, we adjust for layerwise sensitivity and introduce a *progressive compression* strategy, which is supported by our theoretical analysis on how compression errors accumulate in deep networks. Our method is designed to function without model tuning in upcycling stages or task-specific profiling in test stages. Extensive experiments with LLaMA models ranging from 8B to 70B parameters across various tasks show that our approach significantly reduces the GPU memory footprint while maintaining performance.

## 1 Introduction

Autoregressive large language models (LLMs) such as GPT (Achiam et al., 2023), PaLM (Chowdhery et al., 2023), and LLaMA (Touvron et al., 2023), built upon transformer architectures (Vaswani et al., 2017), have shown remarkable capabilities across a wide range of tasks. However, the attention mechanism underpinning those models poses significant challenges to the efficiency of their deployment, particularly the management of the Key-Value (KV) cache. The KV cache is originally designed to accelerate the generation process by storing intermediate attention KV vectors, thus avoiding recomputation of shared prefixes for each autoregressively generated token. Despite reducing computational overhead, the KV cache significantly increases memory footprints, as its size scales linearly with both sequence length and batch size. This drives the need for KV cache compression to enable cost-effective deployment of LLMs across various devices and platforms.

To address the overhead of the original attention mechanism, one prominent line of work aims to design more efficient attention variants, such as multi-query attention (MQA) (Shazeer, 2019) and group-query attention (GQA) (Ainslie et al., 2023), which inherently reduce the corresponding KV cache. Nevertheless, those techniques typically require upcycling existing models. Without proper training, their direct application often results in degraded performance (Ribar et al., 2023; Ainslie et al., 2023; Liu et al., 2024b), thereby making them unsuitable for deployment in resource-

constrained environments. Recently, Liu et al. (2024a) design a multi-head latent attention (MLA) for efficient inference, utilizing low-rank key-value union compression to reduce KV cache. However, similar to MQA and GQA, MLA is also integrated during the model's training cycle, thus not directly applicable to pre-trained LLMs.

In contrast, another line of work focuses on KV cache compression at test time, primarily achieved by dropping tokens while leaving the backbone model intact. Several works design the token eviction policy based on accumulated attention scores(Sheng et al., 2023; Zhang et al., 2024b; Liu et al., 2024b), or heuristics such as special tokens or and relative distance between tokens (Ge et al., 2023) However, these methods either ignore the inter-layer dependency or require attention pattern analysis, and the resulting eviction policy can be task-specific.

In this paper, we propose to compress KV cache from an orthogonal perspective, *i.e.,* the KV weight matrices. As the KV weight matrices are typically characterized by low-rank properties, we perform a *low-rank approximation* to reduce their dimension and thus compress the resulting KV cache. Recognizing that compressed KV caches inevitably introduce information loss to subsequent layers, and that sensitivity to input changes varies across layers, we introduce a *progressive* compression strategy. This approach is grounded in the calculation of cumulative condition numbers for KV weight matrices across different layers, reflecting their sensitivity and guiding the compression strategy. Theoretically, we derive error bounds for both individual layer compression and error propagation through the network. These theoretical results reveal that errors introduced in earlier (shallower) layers are amplified more significantly than those in deeper layers, and informs our progressive compression strategy.

Our method is designed for straightforward implementation, requiring neither model profiling nor detailed inspection of the attention structure. It can be directly applied to pre-trained LLMs by extracting weight matrices and leveraging their inherent properties to swiftly determine optimal layer-wise compression. This approach offers a practical and efficient solution for enhancing LLM performance in memory-constrained deployment scenarios, without the need for model retraining or complex eviction strategy composition.

We evaluate our method on 8B, 13B, and 70B LLaMA models that built upon multi-query attention or group-query attention. Experiments across tasks such as commonsense reasoning, reading comprehension, text summarization, and mathematical reasoning, demonstrate that our approach can reduce substantial GPU memory footprint while maintaining minimal impact on performance.

## 2 Related Works

### 2.1 Attention Mechanism

Attention mechanisms in Transformer models have evolved to enhance efficiency and effectiveness (Vaswani et al., 2017). Multi-Query Attention (MQA)(Shazeer, 2019) reduces memory requirements during decoding, while Grouped-Query Attention (GQA) (Ainslie et al., 2023) balances efficiency and performance by sharing key and value heads among query groups. Recently, Liu et al. (2024a) introduced Multi-head Latent Attention (MLA), using low-rank key-value union compression to optimize inference. However, these approaches are typically integrated during model training, limiting their applicability to pre-trained LLMs. Parallel research efforts have targeted inference efficiency improvements. For example, Pope et al. (2023) developed multi-dimensional partitioning techniques, and de Jong et al. (2022) optimized the Fusion-in-Decoder (FiD) approach (Izacard & Grave, 2020) for more efficient inference. Holmes et al. (2024) introduces SplitFuse which leverages dynamic prompt and generation decomposition and unification to further improve continuous batching and system throughput. In this paper, we contribute to this line of research by improving inference efficiency through the compression of KV cache. Our approach leverages the low-rank property of the attention weight matrices, offering a plug-and-play method to reduce the memory footprint of LLMs during inference without requiring model retraining.

### 2.2 KV Cache Compression

As Large Language Models (LLMs) continue to grow in size and complexity, efficient management of their memory usage during inference has become a critical challenge. Early efforts to compress token hidden states (Guan et al., 2022; Sun et al., 2022; Zhou et al., 2020) are limited to non-

autoregressive models and require retraining, thus motivating research into pruning tokens in the KV cache of auto-regressive LLMs. For instance, Mu et al. (2024) learns to compress prompts into a few special tokens to reduce memory pressure during caching, but this token prediction requires model retraining and could be an expensive overhead during inference. Several methods design token eviction policies based on accumulated attention scores (Sheng et al., 2023; Zhang et al., 2024b; Liu et al., 2024b), or heuristics such as special tokens and relative distance between tokens (Ge et al., 2023). However, these approaches often overlook inter-layer dependencies, potentially resulting in task-specific eviction policies that may not generalize well across different applications. In contrast to token-dropping methods, our study takes a different tack. We focus on compressing the KV cache from the perspective of weight matrix dimension reduction. Importantly, our progressive compression strategy carefully addresses the issue of error propagation across compressed layers, a consideration often ignored in previous methods.

A few studies have explored customized cache budgets across different layers in the context of token dropping, yet no definitive consensus has been reached on the most effective strategies. Zhang et al. (2024a) suggest increasing compression intensity in higher layers based on the assumption that these layers contain less critical information. Conversely, Liu et al. (2024b) argue that significant tokens exhibit greater variability at higher layers, thus larger caches are required to reduce cache misses. While these approaches demonstrate understanding of layer-specific requirements, they depend heavily on task-specific attention patterns. Our approach diverges fundamentally by adopting an orthogonal perspective to compression, focusing on weight matrix dimension reduction rather than token eviction. This approach enables us to establish error propagation bounds across the network and to guide our progressive compression strategy effectively. It eliminates the need to analyze attention patterns for eviction policy design, simplifying implementation and enhancing general applicability across different LLMs.

Concurrently, Liu et al. (2024a) and Yu et al. (2024) modify attention mechanisms to manage KV caches more efficiently during inference. While these methods align with our philosophy of altering attention dynamics, they require either pretraining adjustments or extensive model finetuning to accommodate the modified attention schemas, limiting their practicality in deployed systems. In contrast, our method requires no such training or fine-tuning, offering a plug-and-play solution that seamlessly integrates with pre-trained models to deliver efficient compression without compromising the model's integrity or performance.

# 3 Preliminary: Attention Mechanism and KV Cache

Transformer-based language models use self-attention to weigh the importance of different tokens, thus allowing for the model to focus on different parts of the input sequence. Given an input $X \in \mathbb{R}^{N \times D}$, where $N$ is the sequence length and $D$ is the dimensionality of each token's embedding, we compute the Query ($Q$), Key ($K$), and Value ($V$) matrices by multiplying $X$ with their respective weight matrices: $Q = XW_q, K = XW_k, V = XW_v$.

Then the attention mechanism is as follows:

$$\text{Attention}(Q, K, V) = \text{softmax}\left(\frac{QK^\top}{\sqrt{d_k}}\right) V. \tag{1}$$

Multi-head attention allows the model to jointly attend to information from different representation subspaces at different positions

$$\text{MultiHead}(Q, K, V) = \text{Concat}(\text{head}^1, \ldots, \text{head}^h)W_o, \tag{2}$$

where

$$\text{head}^i = \text{Attention}(X(W_q^i)^T, X(W_k^i)^T, X(W_v^i)^T).^1 \tag{3}$$

Here, $W_q^i$, $W_k^i$, and $W_v^i$ are the weight matrices for the $i$-th attention head, and $W_o$ is the weight matrix for the output linear transformation.

In autoregressive transformers, the computation of attention scales quadratically (*i.e.*, $\mathcal{O}(N^2)$) with the sequence length $N$, as every token in the sequence computes interactions with every other token.

---

[1]This formulation with transposed weight matrices aligns with the implementation found in the models examined in our study. Mathematically, this is equivalent to the standard formulation without transpose. The choice of which form to use depends on implementation details and computational optimizations.

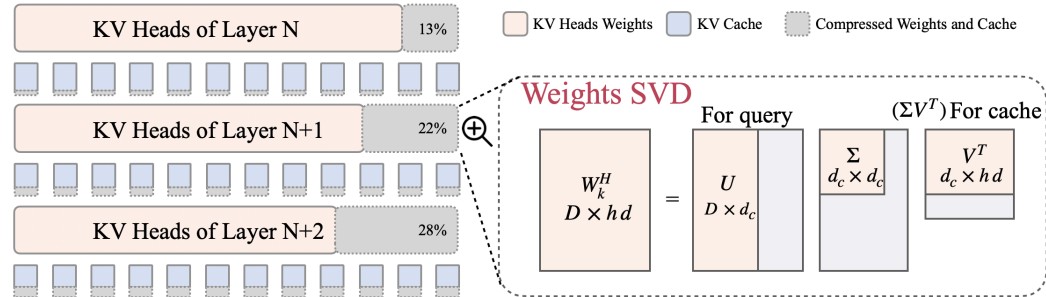

Figure 1: LoRC compresses KV-cache by decomposing the KV weight matrices in attention heads. The progressive compression strategy retains more dimension for KV weights in shallow layers and compresses the KV weights in deep layers more aggressively.

Such scaling is impractical for very large inputs or real-time applications, where speed and efficiency are crucial.

To address this computational bottleneck, KV caches store the results of previous computations of the KV matrices. When processing subsequent tokens, the model can retrieve keys and values from the cache rather than recomputing them, thereby reducing the number of operations to a linear scale with respect to the sequence length. This method trades off increased memory usage for a reduction in computational overhead. The size of KV cache per layer is defined as below:

$$C_{k,v} = b \times N \times h \times d, \tag{4}$$

where $b$ is the batch size, $N$ is the max sequence length in the batch, $h$ is the number of K/V head and $d$ is the head dimension. This linear relationship between cache size and sequence length, as well as batch size, underscores the critical need for efficient compression methods. As described, existing works that can reduce KV cache consumption either require expensive model training in upcycling stages or empirical token eviction policy design at test time. In the following section, we present a novel method for KV cache compression from the perspective of low-rank weight approximation.

## 4 Method

We structure this section as follows. In Section 4.1, we detail the process of compressing the KV cache for a single layer using Singular Value Decomposition (SVD) on weight matrices. Section 4.2 introduces our progressive compression strategy, which determines adaptive compression dimensions for each layer. Finally, Section 4.3 covers additional considerations for handling various attention mechanisms, and Section C addresses the implementation details specific to the rotary position embedding. Figure 1 presents an overview of our method, illustrating the low-rank approximation of the weight matrix and the progressive compression strategy across layers.

### 4.1 KV Cache Compression via Low-rank Approximation of Weight Matrices

Unlike previous approaches that focus on token-level eviction strategies or require model retraining, we propose a novel method that operates at the weight matrix level in the attention mechanism. This approach leverages the inherent low-rank properties of these matrices (as shown in Appendix B), allowing for significant compression without the need for complex token selection algorithms or time-consuming model tuning. By applying a low-rank approximation to the weight matrices, we effectively reduce the dimensionality of the KV cache while preserving the essential information flow through the network.

**Key Matrix Compression:** Figure 1 presents how we implement SVD on the key weight matrices. Specifically, for the $i$-th head in the MHA attention, we decompose its key matrix $W_k^i \in \mathbb{R}^{D \times d}$ to:

$$\text{SVD}(W_k^i)_{D \times d} = U_{D \times d_c} \Sigma_{d_c \times d_c} V_{d_c \times d}^T = U_{D \times d_c} (\Sigma V^T)_{d_c \times d}. \tag{5}$$

For MHA, there are $h$ attention heads, then the decomposition becomes:

$$\text{SVD}(W_k^H)_{D \times hd} = U_{D \times d_c}(\Sigma V^T)_{d_c \times hd} = U_{D \times d_c} \begin{bmatrix} (A^1)_{d_c \times d} & (A^2)_{d_c \times d} & \cdots & (A^h)_{d_c \times d} \end{bmatrix}, \tag{6}$$

where $(A^i)_{d_c \times d}$ is the $i$-th block in the matrix $(\Sigma V^T)_{d_c \times hd}$.

Now we have decomposed the key matrix $W_k^i$ to the multiplication of $U_{D \times d_c}$ and $(\Sigma V^T)_{d_c \times hd}$. We will multiply $X$ with $(\Sigma V^T)_{hd \times d_c}^T$ as the compressed key, which is stored in the KV cache. Through this implementation, we effectively update the size of key cache from $hd$ to $d_c$, where $d_c$ is smaller than $hd$, reducing the memory footprint while keeping the essential information intact.

For $U_{D \times d_c}$, we incorporate it to the query calculation by updating the original query matrix $W_q^H \in \mathbb{R}^{D \times hd}$ as follows:

$$W_{q'}^H = (W_q^H)_{D \times hd} U_{D \times d_c}. \tag{7}$$

Note that the embedding dimension $D$ is equal to the product of the number of attention heads $h$ and the dimension per head $d$, i.e., $D = hd$. Consequently, the updated query matrix $W_{q'}^H \in \mathbb{R}^{D \times d_c}$.

**Value Matrix Compression:** The decomposition for the value matrix follows a similar structure to that of the key matrix, with the difference that we integrate its left singular vectors to the output matrix $W_o$. Specifically, the value matrix is decomposed as:

$$\text{SVD}(W_v^H)_{D \times hd} = U_{D \times d_c}(\Sigma V^T)_{d_c \times hd} = U_{D \times d_c} \begin{bmatrix} (B^1)_{d_c \times d} & (B^2)_{d_c \times d} & \cdots & (B^h)_{d_c \times d} \end{bmatrix} \tag{8}$$

where $(B^i)_{d_c \times d}$ is the $i$-th block in the matrix $(\Sigma V^T)_{d_c \times hd}$. After multiplication with $X$, the dimension of the value cache shrinks from $hd$ to $d_c$, thus reducing memory consumption.

In contrast to the key matrix operation, we incorporate $U_{D \times d_c}$ to the output matrix. To achieve this, we update the output matrix $W_o \in \mathbb{R}^{D \times D}$ as follows:

$$W_{o'} = (U^\top)_{d_c \times D}(W_o)_{D \times D}, \tag{9}$$

resulting in an updated output matrix $W_{o'} \in \mathbb{R}^{d_c \times D}$.

**Compression Ratio:** The compression strategy effectively reduces the dimensions from $N \times d \times h$ for both keys and values to $N \times d_c$, ensuring data integrity and minimizing overhead. This results in a layer compression ratio $\rho = \frac{d_c}{h \times d}$, which quantifies the extent of the reduction.

## 4.2 Progressive Compression Strategy

---
**Algorithm 1** LoRC Algorithm

---
**Require:** Pre-trained LLM with $L$ layers
1: **Initialize** cumulative condition numbers $\tilde{\kappa}_l$
2: **for** $l = L$ to 1 **do**
3:      Compute $\kappa(W_k^l)$ and $\kappa(W_v^l)$
4:      $\tilde{\kappa}_l \leftarrow \prod_{j=l}^{L} \kappa(W_k^j) \cdot \kappa(W_v^j)$
5: **end for**
6: **for** $l = 1$ to $L$ **do**
7:      $d_c^l \leftarrow$ Calculate by Eq. 13
8:      **if** $\tilde{\kappa}_l >$ threshold **then**
9:          Skip compression for layer $l$
10:          **continue**
11:      **end if**
12:      **Key Matrix Compression:**
13:          Perform SVD: $W_k^l = U_k \Sigma_k (V_k^T)$
14:          $\tilde{W}_k^l \leftarrow U_k[:, :d_c^l](\Sigma_k V_k^T)[:d_c^l, :]$
15:          $W_{q'}^l \leftarrow W_q^l U_k[:, :d_c^l]$
16:      **Value Matrix Compression:**
17:          Perform SVD: $W_v^l = U_v \Sigma_v (V_v^T)$
18:          $\tilde{W}_v^l \leftarrow U_v[:, :d_c^l](\Sigma_v V_v^T)[:d_c^l, :]$
19:          $W_{o'}^l \leftarrow U_v[:, :d_c^l]^T W_o^l$
20:      Update KV cache size for layer $l$
21: **end for**

---

Having established low-rank approximation for compressing weight matrices, we now address its dynamic application across network layers. This approach is necessary due to the varying sensitivity of different layers, which significantly affects overall model efficacy and efficiency.

To tackle this challenge, we propose a *progressive* compression strategy for our low-rank approximation of KV weight matrices. Our intuition is that the compressed shallow layers could lead to cascading errors that propagate and amplify through the network. Therefore, we measure the layer sensitivity by the condition numbers of KV matrices to determine *layer-wise* compression dimensions. This approach accounts for each layer's sensitivity to perturbations caused by previously compressed layers, ensuring output variations remain within acceptable ranges. This progressive nature allows for more conservative compression in shallow layers and more aggressive compression in deeper layers, minimizing the risk of error accumulation throughout the network. By carefully balancing compression across layers, we maintain model integrity while achieving significant memory savings.

**Condition Number and Sensitivity Analysis** To ensure that the change in the output $\mathbf{b}_l = \mathbf{A}_l \mathbf{x}_l$ remains within a specified range when the input $\mathbf{x}_l$ changes due to compression in previous layers, we need to consider the sensitivity of the output to such changes. Given a weight matrix $\mathbf{A}_l$, its condition number plays a crucial role in determining the allowable change in $\mathbf{x}_l$. The condition number $\kappa(\mathbf{A}_l)$ is defined as:

$$\kappa(\mathbf{A}_l) = |\mathbf{A}_l|_2 \cdot |\mathbf{A}_l^{-1}|_2 = \frac{\sigma_{\max}(\mathbf{A}_l)}{\sigma_{\min}(\mathbf{A}_l)}, \tag{10}$$

where $\sigma_{\max}(\mathbf{A}l)$ and $\sigma_{\min}(\mathbf{A}_l)$ are the largest and smallest singular values of $\mathbf{A}_l$, respectively. To keep the relative change in the output $\mathbf{b}_l$ within a tolerance $\epsilon$, we utilize the standard definition of the condition number to relate it to the allowable relative change in the input $\mathbf{x}_l$:

$$\frac{|\Delta \mathbf{b}_l|_2}{|\mathbf{b}_l|_2} \le \kappa(\mathbf{A}_l) \cdot \frac{|\Delta \mathbf{x}_l|_2}{|\mathbf{x}_l|_2} \le \epsilon. \tag{11}$$

Solving for the allowable relative change in $\mathbf{x}_l$, we obtain: $\frac{|\Delta \mathbf{x}_l|_2}{|\mathbf{x}_l|_2} \le \frac{\epsilon}{\kappa(\mathbf{A}_l)}$. This inequality indicates that the acceptable change in the input $\mathbf{x}_l$ is *inversely proportional* to the condition number $\kappa(\mathbf{A}_l)$ of the layer's weight matrix. Layers with higher condition numbers are more sensitive to input perturbations, requiring smaller changes in $\mathbf{x}_l$ to maintain the output within the desired range. Given the multi-layer structure of transformers, it is essential to consider not just the condition number of a single layer but the cumulative effect of condition numbers from all preceding layers. This cumulative measure gives a more holistic view of how perturbations might propagate and amplify as data passes through successive layers.

**Cumulative Condition Number**: To effectively manage this across the network, we calculate the cumulative condition number as an estimated layer sensitivity, which we then use to derive the compression dimension. For a model with $L$ layers, we calculate the cumulative condition number for each layer $l$ by multiplying the condition numbers of the current layer and all subsequent layers:

$$\tilde{\kappa}_l = \prod_{j=l}^{L} \kappa(W_k^j) \cdot \kappa(W_v^j), \tag{12}$$

where $W_k^j$ and $W_v^j$ denote the key and value weight matrices of the $j$-th layer, respectively. This cumulative condition number $\tilde{\kappa}_l$ reflects the total amplification of input perturbations from current layer to the final output layer, encompassing the effects of layers from $l$ to $L$.

**Compression Dimension**: Based on the cumulative condition number, we then adjust the compression dimensions for each layer to balance the fidelity and compression rate. More sensitive layers (those with higher cumulative condition numbers) will have less aggressive compression to preserve information, whereas layers with lower sensitivity can be compressed more substantially without significantly affecting the overall network performance. We compute the compressed dimension $d_c^l$ for each layer by scaling $\tilde{\kappa}_l$ using the following function:

$$d_c^l = d_{\max} \times \left[ 1 - \left( \frac{\max_{i \in [1:L]} \log(\tilde{\kappa}_i) - \log(\tilde{\kappa}_l)}{\max_{i \in [1:L]} \log(\tilde{\kappa}_i) - \min_{i \in [1:L]} \log(\tilde{\kappa}_i)} \right) \times \left( 1 - \frac{d_{\min}}{d_{\max}} \right) \right], \tag{13}$$

where $d_{\max}$ is the maximum allowable compressed dimension, and $d_{\min}$ is the minimum one. The logarithmic scale mitigates the effect of large variations in the cumulative condition numbers, providing a more balanced sensitivity metric across layers. This equation ensures that layers with higher sensitivity (larger $\tilde{\kappa}_l$) retain more dimensions (larger $d_l$), while less sensitive layers can be compressed more aggressively.

## 4.3 Multi-head Attention and Group-query Attention

The above derivation in Section 4.1 holds for standard MHA, where the model dimension $D$ equals to the multiplication of number of head and head dimension $h \times d$. For GQA, the number of KV heads is reduced as shown in Table 3. To adapt such implementation, we can still follow the above procedure for cache compression. After fetching the key and value from cache, we just need to repeat them according to the number of the total attention heads.

## 5 Error Bounds for KV Cache Compression

In this section, we derive error bounds for our KV cache compression method, considering both individual layer errors and their propagation through a deep network. These theoretical results provide insights into how the matrix decomposition-based compression affects the network's performance and guide the progressive compression strategy to balance model efficiency and performance.

### 5.1 Error Bound for Key/Value Matrix Approximation

**Theorem 1** *Let $W \in \mathbb{R}^{m \times n}$ be a weight matrix (either key or value), and let $\tilde{W} \in \mathbb{R}^{m \times n}$ be its rank-$k$ approximation obtained via truncated singular value decomposition (SVD). For any input vector $x \in \mathbb{R}^n$, the error introduced by the approximation is bounded by:*

$$\|Wx - \tilde{W}x\|_2 \leq \sigma_{k+1}\|x\|_2, \tag{14}$$

*where $\sigma_{k+1}$ is the $(k+1)$-th singular value of $W$.*

The proof is provided in Appendix A.1. This theorem quantifies the error introduced at a single layer due to compressing the weight matrix. The bound indicates that the error is directly proportional to the $(k+1)$-th singular value of $W$ and the norm of the input vector $x$. Larger singular values correspond to directions of significant variance in the data, so truncating smaller singular values (which represent less significant features) minimizes the error introduced by compression.

### 5.2 Single Layer Error Bound Including Nonlinearities

We now extend the analysis to include the effect of nonlinearities within a single layer. We derive an error bound that accounts for both the approximation of the weight matrix and the layer's nonlinear activation function. For simplicity, we analyze the error introduced by compressing each weight matrix (key or value) individually.

**Theorem 2** *Consider a single layer applying a linear transformation $W$ followed by a nonlinearity $\phi$ with Lipschitz constant $L_\phi$. Let $\tilde{W}$ be the compressed version of $W$ obtained via truncated SVD with rank $k$. For any input vector $x \in \mathbb{R}^n$, the error at the output of the layer is bounded by:*

$$\left|\phi(Wx) - \phi(\tilde{W}x)\right| \leq L_\phi \sigma_{k+1}\|x\|_2. \tag{15}$$

The proof is straightforward by using Theorem 1 and the Lipschitz property of $\phi$, we present it as the base case in the proof of Theorem 3, which is detailed in Appendix A.2.

This theorem shows that the error introduced by the compressed weight matrix propagates through the nonlinearity, scaled by the Lipschitz constant of the activation function. While considering both matrices simultaneously complicates the bounds due to their interactions within the attention mechanism, it is still feasible to derive combined error bounds because the attention mechanism allows us to mathematically bound these interactions. The total error due to simultaneous compression can be bounded by the sum of their individual approximation errors, scaled by a constant. However, for simplicity and clarity in the following derivation, we use the simplified version that considers each matrix individually.

### 5.3 Error Propagation Bound

**Theorem 3** *Consider an $L$-layer network where each layer $i$ applies a linear transformation $W_i$ followed by a nonlinearity $\phi$ with Lipschitz constant $L_\phi$. Let $\tilde{W}_i$ be the compressed version of $W_i$ obtained via truncated SVD with rank $k_i$. The error at the output of the network is bounded by:*

$$\|x_L - \tilde{x}_L\|_2 \leq \sum_{i=1}^{L} \left( \sigma_{k_i+1}^{(i)} L_\phi^{L-i} \prod_{j=i+1}^{L} \|W_j\|_2 \right), \tag{16}$$

where $x_L$ and $\tilde{x}_L$ are the outputs of the original and compressed networks, respectively; $\sigma_{k_i+1}^{(i)}$ is the $(k_i + 1)$-th singular value of $W_i$; $\|W_j\|_2$ denotes the spectral norm of $W_j$; and $L_\phi$ is the Lipschitz constant of the activation function $\phi$.

We detail the proof in Appendix A.2. Until now, we have established an upper bound on the cumulative error at the network's output due to compression of weight matrices across multiple layers. It is important to note that the nonlinearities characterized by the Lipschitz constant $L_\phi$ represent a simplification. In practice, transformer models like LLaMA incorporate complex nonlinear components, so the exact error propagation may deviate from this simplified bound due to intricate nonlinearities. Despite these complexities, the theorem still offers insights into how compression errors may accumulate in deep networks. Specifically, it reveals that errors introduced in earlier (shallower) layers are amplified more significantly than those in deeper layers because they pass through more subsequent transformations and nonlinearities.

This understanding supports our design of a progressive compression strategy, where we compress shallow layers less aggressively than deeper ones. By preserving more information in the early layers (i.e., retaining more singular values), we minimize the initial errors that could be significantly amplified throughout the network. This approach helps maintain overall model performance while still achieving substantial compression in deeper layers, where the impact on the final output is less pronounced due to reduced error amplification.

# 6 Experiment

## 6.1 Models

We conduct experiments using two attention mechanisms, Multi-Head Attention (MHA) (Vaswani et al., 2017) and Graph Query Attention (GQA) (Ainslie et al., 2023), across three models: LLaMA-2-13B, LLaMA-3-Instruct-8B, and LLaMA-3-Instruct-70B. The LLaMA-2 family incorporates the MHA mechanism, while the LLaMA-3 family is based on the GQA framework. We list the model specifications in Table 3. Note that for the models based on MHA, the number of KV heads is equal to the number of attention heads, so the weight matrices of KV are square matrices. The models based on GQA use an intermediate number of key-value heads to group the query heads, with an adjustment on the shape of KV weight matrices.

## 6.2 Implementation Details

In practice, we set thresholds to exclude compression on layers with high cumulative condition numbers: 30 for LLaMA-3-Instruct-8B, and 90 for LLaMA-2-13B and LLaMA-3-Instruct-70B. The $d_{max}$ equals to the original head dimension, while $d_{min}$ varies based on the target compression ratio. For baseline methods, we have the same refrained layers while applying the uniform compression ratios across compressed layers instead of using a progressive compression strategy.

## 6.3 Dataset

We follow Touvron et al. (2023) to evaluate our methods on the following tasks: **BoolQ** (Clark et al., 2019) for reading comprehension, **XSum** (Narayan et al., 2018) for text summarization. **Openbook QA** (Mihaylov et al., 2018) for commonsense reasoning, and **GSM8K** (Cobbe et al., 2021) for mathematical reasoning. We use ROUGE score (Lin, 2004) as the evaluation metric for XSum and accuracy for the other tasks. We report 2-shot results for LLaMA-2 models on BoolQ, and 0-shot results for other settings.

## 6.4 Main Results

Figure 2 presents our main results on four datasets with different KV cache budgets. Compared to the full-cache model, LoRC achieves on-par performance with a significant compression ratio, and the performance degradation is still nearly negligible with a $60\%$ compression ratio on most datasets. When slightly compressed, LoRC could even enhance model performance in some cases. Note that our method requires no model training or model profiling, the only efforts are SVD on weight matrices which requires minimal computational cost compared to the LLM inference. Such plug-and-play

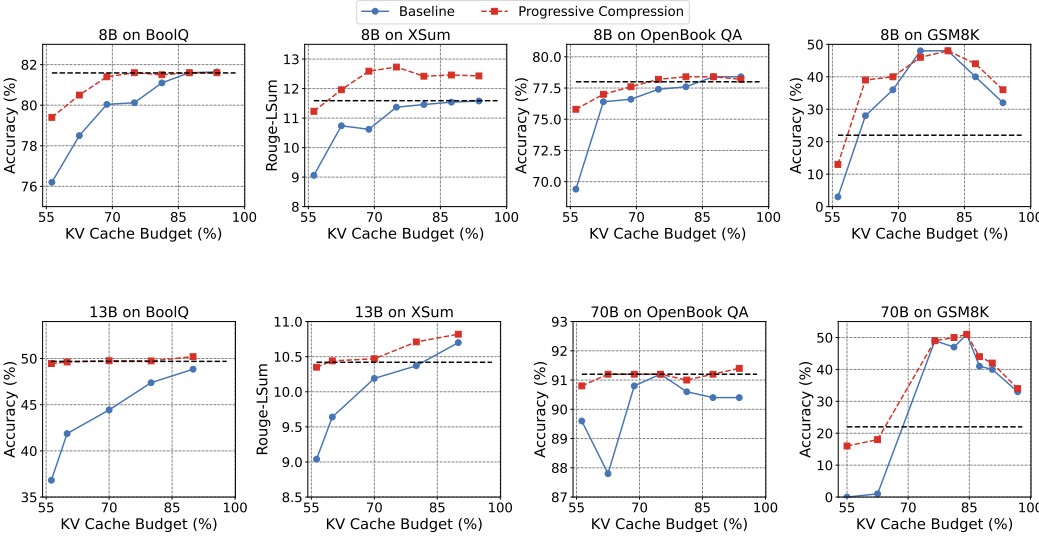

Figure 2: Performance of KV cache compression on LLaMA models. LoRC compresses the KV weights with a progressive strategy, while the baselines compress each layer with the same ratio. The horizontal dashed line indicates the performance with a full-cache model.

merits make our method easily integrable in resource-constrained environments, enabling efficient model deployment with limited KV cache budgets.

In Figure 2, one interesting observation is that in some cases the model with a compressed KV cache leads to better performance. Particularly, on the GSM8K dataset, performing KV cache compression leads to more than 10% performance improvement. This phenomenon aligns with findings reported in the literature (Ge et al., 2023). Also, similar effects have been documented in the context of improving reasoning by applying low-rank decomposition on the MLP layers (Sharma et al., 2023). We believe this phenomenon demonstrates the feasibility of conducting task-specific profiling for better performance, or adapting our proposed method in model finetuning.

## 6.5 Single Layer Profiling

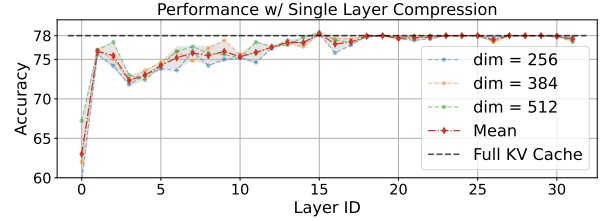

Figure 3: Single-layer compression results. This experiment uses LLaMA-3-Instruct-8B on the OpenBookQA dataset.

To investigate the impact of compression at different layers, we conduct experiments on single-layer compression as shown in Fig. 3. We use LLaMA-3-Instruct-8B on OpenBook QA for this experiment. The original dimension of the KV head is 1024, and we select compression dimensions from $[256, 384, 512]$ to compress each single layer while keeping all other layers untouched.

Figure 3 shows clear layer-specific variability, indicating that some layers are more susceptible to compression than others, particularly in the shallow layers. It is observed that the deep layers (*i.e.,* layers 15–31 of the 32-layer LLaMA-3-Instruct 8B model), despite the reduction in dimensions, maintain performance closely approaching the full KV Cache baseline. This suggests that these layers can sustain robust performance even when subjected to significant parameter reduction. This finding supports our progressive compression strategy for optimizing model efficiency without significantly compromising the model's effectiveness.

Table 1: Performance comparison between compression on shallow layers and deep layers on OpenBookQA. For our progressive compression strategy, we report the performance at the 60% overall compression ratio. For layer-0 compression and shallow blocks compression, we use a 50% layer compression ratio within the chosen strategy. Hence, the overall compression ratio is 98.44% for the layer-0 compression, and 93.75% for the shallow blocks compression.

| Model | Baseline | Ours | Layer 0 | Shallow Blocks (1/8) |
|---|---|---|---|---|
| LLaMA-2-13b | 76.6 | 77.4 (↑ 0.8) | 77.2 (↑ 0.6) | 74.8 (↓ 1.8) |
| LLaMA-3-Instruct-8b | 78.0 | 77.4 (↓ 0.6) | 67.2 (↓ 10.8) | 61.4 (↓ 16.6) |
| LLaMA-3-Instruct-70b | 91.2 | 91.2 (↑ 0.0) | 84.2 (↓ 7.0) | 23.2 (↓ 68.0) |

## 6.6 Curse of shallow layers

To validate the intuition of the progressive compression strategy that the noise caused by shallow compressed layers will be amplified more after propagation, we compare it to compressing the first layer and the shallow blocks (i.e., the first 1/8 layers in a model) on 3 LLaMA models.

Table 1 shows how the compressed shallow layers impact the model performance, taking the baseline full-cache model and our method as reference. The results indicate that compressing only the first layer can lead to a performance decline, with reductions ranging from minimal to moderate. For instance, the LLaMA-3-70B gives a 7.0% decrease, while the LLaMA-3-Instruct-8b shows a more substantial drop of 10.8%. When compressing the shallow blocks, the impact is more pronounced. The LLaMA-3-Instruct-8B suffers a 16.6% reduction. Notably, the LLaMA-3-Instruct-70b model shows a drastic 68.0% decline, highlighting a significant sensitivity to shallow layer compression.

These findings underscore the importance of careful layer selection in compression strategies and validate the effectiveness of our progressive compression method, as the choice of layer to compress can have a substantial impact on model performance, particularly in larger or more complex models.

## 6.7 Memory footprint reduction analysis

Table 2: Summary of Model Sizes, KV cache usage and performance drop. Experiments were conducted with a batch size of 64 and a sequence length of 2048 for all models.

| Model | KV Cache | | | | | Average Performance Drop |
|---|---|---|---|---|---|---|
| | Full | dim | dim_c | Ours | Compression Ratio | |
| **LLaMA-2-13B** | 50G | 5120 | 2048 | 27.5G | 55% | 0.47% |
| **LLaMA-3-8B** | 8G | 1024 | 512 | 4.8G | 60% | 0.92% |
| **LLaMA-3-70B** | 20G | 1024 | 512 | 11G | 55% | 0.22% |

We report the memory footprint reduction in Table 2. By controlling the performance drop averaged on the four tasks less than 1%, we can achieve a considerable compression ratio from 55%-60%. For the LLaMA-3 models in which the GQA has already been employed to save the KV cache, we further achieve a significant compression ratio. Note that we have excluded the GSM8k results for the performance drop calculation for a fair comparison.

## 7 Conclusions

In conclusion, we proposed LORC, a novel approach to KV cache compression that capitalizes on the inherent low-rank properties of weight matrices. Our method employs a progressive layer-wise compression strategy, implementing a post-hoc low-rank approximation to circumvent the complexities and limitations associated with token-level eviction strategies and model retraining. Moreover, we provide theoretical analysis, deriving error bounds for layer compression and error propagation in deep networks, supporting our design of progressive compression strategy. This theoretically grounded and universally applicable approach preserves model integrity and performance across diverse tasks, attention mechanisms, and model scales. Our comprehensive experimental results demonstrate that LORC significantly reduces GPU memory requirements while minimally impacting performance. This approach offers a robust and efficient solution of KV cache compression, without requiring attention pattern analysis or model tuning.

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

## A Detailed Proofs

### A.1 Proof of Theorem 1

*The proof of Theorem 1 is presented here for completeness.*

**Proof.**

Let $W = U\Sigma V^\top$ be the full SVD of $W$, where $U \in \mathbb{R}^{m \times m}$ and $V \in \mathbb{R}^{n \times n}$ are orthogonal matrices, and $\Sigma = \mathrm{diag}(\sigma_1, \ldots, \sigma_n)$ with singular values $\sigma_1 \geq \sigma_2 \geq \cdots \geq \sigma_n \geq 0$.

The rank-$k$ approximation $\tilde{W}$ is given by:

$$\tilde{W} = U_k \Sigma_k V_k^\top,$$

where $U_k$, $\Sigma_k$, and $V_k$ are truncated versions of $U$, $\Sigma$, and $V$, respectively, keeping only the first $k$ singular values and corresponding vectors.

We have:

$$
\begin{aligned}
\|Wx - \tilde{W}x\|_2 &= \|(W - \tilde{W})x\|_2 \\
&= \|U(\Sigma - \Sigma_k)V^\top x\|_2 \\
&= \|(\Sigma - \Sigma_k)V^\top x\|_2, \quad \text{since } U \text{ is orthogonal} \\
&= \|\mathrm{diag}(0, \ldots, 0, \sigma_{k+1}, \ldots, \sigma_n)V^\top x\|_2 \\
&\leq \sigma_{k+1}\|V^\top x\|_2 \\
&= \sigma_{k+1}\|x\|_2, \quad \text{since } V \text{ is orthogonal.}
\end{aligned}
$$

$\square$

### A.2 Proof of Theorem 3

*We present the proof of Theorem 3, including an adjustment for activation functions with Lipschitz constant $L_\phi$.*

**Proof.**

Let $x_i$ and $\tilde{x}_i$ denote the outputs of the $i$-th layer in the original and compressed networks, respectively. We prove by induction that:

$$\|x_i - \tilde{x}_i\|_2 \leq \sum_{s=1}^{i} \left( \sigma_{k_s+1}^{(s)} L_\phi^{i-s} \prod_{j=s+1}^{i} \|W_j\|_2 \right). \tag{17}$$

**Base Case ($i = 1$).**

Using Theorem 1 and the Lipschitz property of $\phi$:

$$
\begin{aligned}
\|x_1 - \tilde{x}_1\|_2 &= \|\phi(W_1 x_0) - \phi(\tilde{W}_1 x_0)\|_2 \\
&\leq L_\phi \|W_1 x_0 - \tilde{W}_1 x_0\|_2 \\
&\leq L_\phi \sigma_{k_1+1}^{(1)} \|x_0\|_2.
\end{aligned}
$$

**Inductive Step.**

Assume the inductive bound holds for layer $i - 1$. For layer $i$:

$$\|x_i - \tilde{x}_i\|_2 = \|\phi(W_i x_{i-1}) - \phi(\tilde{W}_i \tilde{x}_{i-1})\|_2$$
$$\leq L_\phi \|W_i x_{i-1} - \tilde{W}_i \tilde{x}_{i-1}\|_2$$
$$\leq L_\phi \left( \|W_i(x_{i-1} - \tilde{x}_{i-1})\|_2 + \|(W_i - \tilde{W}_i)\tilde{x}_{i-1}\|_2 \right)$$
$$\leq L_\phi \left( \|W_i\|_2 \|x_{i-1} - \tilde{x}_{i-1}\|_2 + \sigma^{(i)}_{k_i+1} \|\tilde{x}_{i-1}\|_2 \right).$$

We can bound $\|\tilde{x}_{i-1}\|_2$ using the triangle inequality:

$$\|\tilde{x}_{i-1}\|_2 \leq \|x_{i-1}\|_2 + \|x_{i-1} - \tilde{x}_{i-1}\|_2.$$

Assuming that $\|x_{i-1}\|_2$ is bounded (which is reasonable in practice due to normalization techniques), and applying the inductive hypothesis, we can express $\|x_i - \tilde{x}_i\|_2$ in terms of the accumulated errors up to layer $i$.

By recursively applying this inequality and summing over all layers, we obtain the bound stated in Theorem 3.

$\square$

### A.3 Note on Activation Functions and the Lipschitz Constant

It is important to note that Theorem 3 assumes the activation function $\phi$ has a Lipschitz constant $L_\phi$, which reflects how much the function can amplify differences in its input. For activation functions like ReLU, which are 1-Lipschitz, the error bound simplifies and indicates minimal error amplification through the activation layers.

However, the LLaMA model family uses activation functions such as SwiGLU and GELU, whose derivatives can exceed 1, making them not 1-Lipschitz. For networks employing such activation functions, the error propagation bound in Theorem 2 is adjusted by incorporating a Lipschitz constant $L_\phi$, which may be greater than 1. This adjustment accounts for the potential additional error amplification introduced by the activation functions.

## B Reconstruction Error of Matrix SVD

Given a pre-trained LLM, we conduct layer-wise weight matrix decomposition and reconstruction. We found that these matrices are low-rank and therefore can be reconstructed with low-dimension matrices, resulting in minimal reconstruction error. It means instead complex eviction policy design at the token level, we can turn to the attention level to develop a model and task agnostic KV cache compression method. We present the relative reconstruction error in Figure 4, which is computed as below.

```python
# Code for matrix reconstruction error calculation
matrix_reconstructed = torch.mm(torch.mm(U_reduced, S_reduced_diag),
    V_reduced)
error = torch.norm(matrix - matrix_reconstructed, p='fro')
relative_error = error / torch.norm(matrix, p='fro')
```

## C Adjusted Position Embedding

Su et al. (2024) propose a rotary position embedding (RoPE) and it has been used in most recent LLMs. Applying RoPE to self-attention gives

$$q_m^T k_n = (R^d_{\Theta,m} W_q^T x_m)^T (R^d_{\Theta,n} W_k^T x_n) = x^T W_q R^d_{\Theta,n-m} W_k^T x_n, \tag{18}$$

where $\Theta$ is a pre-defined rotary matrix, $m$ and $n$ denotes the token position. In practice, the rotation matrix $R^d_{\Theta,n-m}$ is decomposed as $(R^d_{\Theta,m})^T$ and $R^d_{\Theta,n}$ to rotate the query and key separately, and the KV cache stores the rotated keys. To ensure that our compressed keys are compatible with the

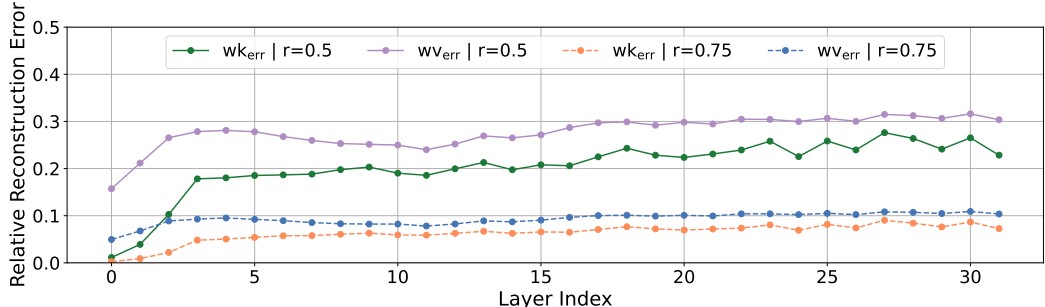

Figure 4: Layerwise relative reconstruction errors. $wk_{err}$ and $wv_{err}$ denote the relative difference between the original key/value matrices and their corresponding low-rank approximations measured using the Frobinus norm. The compression ratio is computed as $r = \frac{d_c}{N_h \times d_h}$, where $N_h$ is the number of attention heads and $d_h, d_c$ is the original and compressed hidden dimensions respectively.

rotary operation, we adjust the position embedding pipeline. Specifically, we store the compressed keys $X(\Sigma V^\top)^\top_{D \times d_c}$ in cache, while incorporating the rotation and key projection into the query computation to streamline the process.

RoPE does bring additional computations to LoRC. To address this, we follow H2O (Zhang et al., 2024b) and FastGen (Ge et al., 2023) to develop a customized kernel that fuses cache reconstruction and rotation operations. This approach minimizes memory transfers and computational overhead, enhancing the overall throughput.

Specifically, we leverage Triton to implement a fused kernel that optimizes memory access and computation. With this kernel, we are able to combine key reconstruction from compressed representation and apply RoPE rotations on-the-fly. It also streamlines memory access because the single data load from global memory for compressed keys requires minimal data movement. In a complete RoPE workflow, there are two stages:

- **Caching:** We project keys to low-rank space without position embedding and store only the compressed representation.

- **Attention computation:** Within the customized kernel, we load compressed keys into shared memory, reconstruct and apply RoPE in a fused operation, and then compute attention scores with position-aware keys.

With these engineering efforts, we achieve higher throughput compared to baseline configurations.

Table 3: Model Architectures.

| Model | Attention | Layers | Heads | KV Heads | Head Dimension | Model Dimension | Weight Shape |
|---|---|---|---|---|---|---|---|
| LLaMA-2-13B | MHA | 40 | 40 | 40 | 128 | 5120 | $5120 \times 5120$ |
| LLaMA-3-Instruct-8B | GQA | 32 | 32 | 8 | 128 | 4096 | $4096 \times 1024$ |
| LLaMA-3-Instruct-70B | GQA | 80 | 64 | 8 | 128 | 8192 | $8192 \times 1024$ |

# D  Latency Analysis

Table 4: Latency analysis for sensitivity calculation and SVD processing.

| Model | Computing Resource | Sensitivity Calculation (s) | SVD Processing (s) |
|---|---|---|---|
| LLaMA-3-8B-Instruct | NVIDIA H100 80GB HBM3 $\times$ 1 | 30.6 | 33.3 |
| LLaMA-3-70B-Instruct | NVIDIA H100 80GB HBM3 $\times$ 8 | 12.2 | 14.3 |

The latency analysis is shown in Table 4. Note that sensitivity calculation only takes place once for a given model, and the SVD processing is a one-time implementation during the model initialization

stage. Such latency will not affect inference speed. Compared to the whole inference duration, the latency incurred by these processes is negligible. For example, the LLaMA-3-70B-Instruct model requires approximately 1 hour and 8 minutes to process 1,000 summaries from the XSUM dataset with a batch size of 32 and a sequence length of 8,000. The combined latency introduced by sensitivity calculations and SVD processing represents only 0.6% of the total inference time.

# E   Throughput Analysis

Our method involves additional computations to recover the compressed cache and manage RoPE. To address this, we developed a customized kernel that fuses cache reconstruction and rotation operations. This approach minimizes memory transfers and computational overhead, enhancing the overall throughput.

Following H2O () we conducted throughput experiments with fixed input and output sequence lengths using the LLaMA-3-70B-Instruct model on a node equipped with eight NVIDIA H100 80GB HBM3 GPUs. The results below indicate that our engineering efforts can streamline the attention computation with LoRC compression, thereby achieving higher throughput compared to full cache scenarios.

Table 5: Throughput of LLaMA-3-70B-Instruct with and without LoRC compression.

| Input Length | Output Length | Batch Size | Full Cache Throughput | LoRC-60% Throughput | Speedup |
|---|---|---|---|---|---|
| 1024 | 2048 | 32 | 52.75 tokens/s | 60.08 tokens/s | × 1.14 |
| 1024 | 4096 | 32 | 78.66 tokens/s | 98.74 tokens/s | × 1.26 |

# F   Comparison with Other KV Cache Compression Methods

We compare LoRC with a token-eviction method – H2O (Zhang et al., 2024b) and a quantization method – KIVI (Liu et al., 2024c) in this section. We conduct experiments using LLaMA-3-8B-Instruct. For H2O and LoRC, we keep the same 60% KV cache budget. For KIVI, we use the KIVI-4-bit implementation. We evaluate accuracy on BoolQ and OpenBook QA, and Rouge-Lsum for XSum. The results below show LoRC can preserve a better performance compared to token-eviction and quantization methods when KV cache is aggressively compressed.

Table 6: Performance comparison across different KV cache compression methods.

| Method | BoolQ | XSum | OpenBookQA |
|---|---|---|---|
| Full Cache | 81.6 | 11.6 | 78.0 |
| H2O (Zhang et al., 2024b) | 76.4 | 10.5 | 75.1 |
| KIVI (Liu et al., 2024c) | 77.6 | 10.3 | 74.8 |
| LoRC | 79.2 | 11.2 | 75.7 |

For throughput comparison with the other methods, we use the metric seconds per iteration (s/it) on XSum. We adhered to the same 60% KV cache budget for both H2O and LoRC. In an effort to demonstrate compatibility with other KV cache compression methods, we integrated LoRC with both H2O and standard 8-bit quantization. Specifically, we allocated a 70% KV cache budget to LoRC and 85% to H2O, resulting in approximately a 60% overall cache budget. A similar configuration was used for the combination of LoRC with 8-bit quantization.

Table 7: Throughput and performance comparison of different methods..

| | Full Cache | H2O | KIVI | LoRC | LoRC w/ H2O | LoRC w/ 8-bit |
|---|---|---|---|---|---|---|
| Performance | 11.6 | 10.5 | 10.3 | 11.2 | 11.0 | 10.9 |
| s/it | 228.5 | 192.7 | 186.6 | 203.4 | 195.9 | 193.7 |

Our results show that LoRC achieves superior performance compared to existing compression methods while maintaining competitive throughput. While the SVD-based computations introduce some overhead, LoRC still has a higher throughput than its full cache baseline because we deployed a customized kernel to streamline the attention computation. It is important to note that our primary objective was not to develop a Pareto-optimal method (simultaneously optimizing both performance and throughput), but rather to introduce an orthogonal approach to KV cache compression and establish the progressive compression strategy.

Additionally, the integration of LoRC with both token-eviction and quantization methods demonstrates its compatibility with existing compression methods. With only a slight performance trade-off (11.0/10.9 vs 11.2), these hybrid configurations achieve better throughput compared to using LoRC alone, which shows the potential of combining LoRC with orthogonal compression approaches.

## G  Enhancements in memory-constrained deployment scenarios

Table 8: LLaMA-3-8B-Instruct model on a single GPU with 24GB memory.

| Max Sequence Length | Batch Size | 8B w/ Full Cache | 8B w/ LoRC-60% |
|---|---|---|---|
| 2048 | 20 | OOM | 22.12 G |

Table 9: LLaMA-3-70B-Instruct model on 8 GPUs with 24GB memory.

| Max Sequence Length | Batch Size | 70B w/ Full Cache | 70B w/ LoRC-60% |
|---|---|---|---|
| 2048 | 8 | OOM | 22.97 G |

LoRC demonstrates significant benefits in memory-constrained deployment scenarios. To further validate this, we provide additional experiments here. These results demonstrate that LoRC enables the deployment of models that would otherwise be impossible to run with full KV cache on consumer GPUs.

## H  Prompts used in experiments

We present our prompts used for different datasets here. We use a few-shot setting for LLaMA-2 models on OpenBookQA (1-shot), BoolQ (2-shot), and GSM8k (8-shot), and zero-shot setting for other experiments.

**OpenBookQA**

```
def format_examples(examples):
    example_prompts = []
    for j in range(1):
        question = examples['question_stem'][j]
        fact = examples['fact1'][j]
        choices = examples['choices'][j]['text']
        labels = examples['choices'][j]['label']
        formatted_choices = "\n".join(f"{label}) {text}" for label,
    text in zip(labels, choices))
        answer = examples['answerKey'][j]
        example_prompt = f"Fact: {fact}\nQuestion: {question}\nOptions
    :\n{formatted_choices}\nAnswer: {answer}\n"
        example_prompts.append(example_prompt)
    return "\n---\n".join(example_prompts)

def create_prompts_from_data(data, example_context):
    prompts = []
    answers = []
    for i in range(len(data['id'])):
```

```
596 19        question = data['question_stem'][i]
597 20        fact = data['fact1'][i]
598 21        choices = data['choices'][i]['text']
599 22        labels = data['choices'][i]['label']
600 23        formatted_choices = "\n".join(f"{label}) {text}" for label,
601    text in zip(labels, choices))
602 24
603 25        task_intro = "You will be provided with a fact and a related
604    question. Your task is to use the given fact to choose the correct
605     answer from the provided options."
606 26        prompt = f"Task Introduction:\n{task_intro}\n1-Shot Examples:\
607    n{example_context}\n---\nFact: {fact}\nQuestion: {question}\
608    nOptions:\n{formatted_choices}\nAnswer:"
609 27        prompts.append(prompt)
610 28        answers.append(data['answerKey'][i])
611 29    return prompts, answers
612 30
613 31
614 32 def extract_option_label(outputs):
615 33    answer_labels = []
616 34    for output in outputs:
617 35        match = re.search(r'\b([A-D])\b', output)
618 36        if match:
619 37            answer_labels.append(match.group(1))
620 38        else:
621 39            answer_labels.append(None)
622 40    return answer_labels
```

**BoolQ**

```
624 1 def get_examples(dataset, num_examples):
625 2    selected_examples = dataset.shuffle(seed=42).select(range(
626    num_examples))
627 3    examples = []
628 4    for i in range(num_examples):
629 5        passage = selected_examples['passage'][i]
630 6        question = selected_examples['question'][i]
631 7        answer = "yes" if selected_examples['answer'][i] else "no"
632 8        examples.append((passage, question, answer))
633 9
634 10    example_section = "\n\n".join([
635 11        f"Example {i + 1}:\nPassage: {ex[0]}\nQuestion: {ex[1]}\
636    nAnswer: {ex[2]}" for i, ex in enumerate(examples)
637 12    ])
638 13    return example_section
639 14
640 15 def create_prompts_from_data(data, example_section=None):
641 16    task_description = "For each passage and question, determine if
642    the answer to the question is 'yes' or 'no' based on the passage
643    provided."
644 17
645 18    prompts = []
646 19    references = []
647 20    for question, passage, answer in zip(data['question'], data['
648    passage'], data['answer']):
649 21        prompt = f"{task_description}\n\n2-Shot Examples:{
650    example_section}\n\nPassage: {passage}\nQuestion: {question}\n\
651    nAnswer (yes or no):"
652 22        prompts.append(prompt)
653 23        references.append("yes" if answer else "no")
654 24    return prompts, references
655 25
656 26 def extract_answer(generated_text: str) -> str:
657 27    normalized_text = generated_text.lower().strip()
658 28    if normalized_text.startswith("yes"):
```

```
29        return "yes"
30    elif normalized_text.startswith("no"):
31        return "no"
32    return "unknown"
```

**XSum**

```
1 def create_prompts_from_data(data):
2     prompts = []
3     references = []
4     for article, summary in zip(data['document'], data['summary']):
5         prompt = f"Provide a concise summary of the text below: {
      article}\n\nSummary:"
6         prompts.append(prompt)
7         references.append(summary)
8     return prompts, references
```

**GSM8k**

```
1 def create_prompts_from_data(data, examples):
2     content = f"Please give a step-by-step answer to the question. You
       have to put your final numeric answer at the end, without any
      extra sign, prefix, or suffix, just pure integer numbers, in the
      format: \n#### answer\n Done, make sure to separate the final
      numeric answer with \n####"
3
4     prompts = []
5     references = []
6
7     example_section = ""
8     for ex_question, ex_answer in examples:
9         example_section += f"\nExample Question: {ex_question}\
      nExample Answer: {ex_answer}\n"
10
11     for question, answer in zip(data['question'], data['answer']):
12         prompt = f"{example_section}\nQuestion: {question}\n{content}.
      "
13         prompts.append(prompt)
14         _, extracted_answer = extract_answer(answer)
15         references.append(extracted_answer)
16     return prompts, references
17
18 def extract_answer(completion):
19     start_idx = completion.find("####")
20     if start_idx == -1:
21         return completion, 'None'
22     start_idx += 4  # Move past '####'
23     end_idx = completion.find('\n', start_idx)
24     if end_idx == -1:
25         end_idx = len(completion)
26     answer = completion[start_idx:end_idx].strip()
27     return completion[:end_idx], answer
28
29
30 def calculate_accuracy(predictions, references):
31     correct = sum([1 for (_, pred), ref in zip(predictions, references
      ) if pred.lower() == ref.lower()])
32     return correct, len(predictions)
```

# I  Throughput Analysis

Our method involves additional computations to recover the compressed cache and manage RoPE.
To address this, we developed a customized kernel that fuses cache reconstruction and rotation

operations. This approach minimizes memory transfers and computational overhead, enhancing the overall throughput.

Following H2O () we conducted throughput experiments with fixed input and output sequence lengths using the LLaMA-3-70B-Instruct model on a node equipped with eight NVIDIA H100 80GB HBM3 GPUs. The results below indicate that our engineering efforts can streamline the attention computation with LoRC compression, thereby achieving higher throughput compared to full cache scenarios.

Table 10: Throughput of LLaMA-3-70B-Instruct with and without LoRC compression.

| Input Length | Output Length | Batch Size | Full Cache Throughput | LoRC-60% Throughput | Speedup |
|---|---|---|---|---|---|
| 1024 | 2048 | 32 | 52.75 tokens/s | 60.08 tokens/s | × 1.14 |
| 1024 | 4096 | 32 | 78.66 tokens/s | 98.74 tokens/s | × 1.26 |

# J  Comparison with Other KV Cache Compression Methods

We compare LoRC with a token-eviction method – H2O (Zhang et al., 2024b) and a quantization method – KIVI (Liu et al., 2024c) in this section. We conduct experiments using LLaMA-3-8B-Instruct. For H2O and LoRC, we keep the same 60% KV cache budget. For KIVI, we use the KIVI-4-bit implementation. We evaluate accuracy on BoolQ and OpenBook QA, and Rouge-Lsum for XSum. The results below show LoRC can preserve a better performance compared to token-eviction and quantization methods when KV cache is aggressively compressed.

Table 11: Performance comparison across different KV cache compression methods.

| Method | BoolQ | XSum | OpenBookQA |
|---|---|---|---|
| Full Cache | 81.6 | 11.6 | 78.0 |
| H2O (Zhang et al., 2024b) | 76.4 | 10.5 | 75.1 |
| KIVI (Liu et al., 2024c) | 77.6 | 10.3 | 74.8 |
| LoRC | 79.2 | 11.2 | 75.7 |

For throughput comparison with the other methods, we use the metric seconds per iteration (s/it) on XSum. We adhered to the same 60% KV cache budget for both H2O and LoRC. In an effort to demonstrate compatibility with other KV cache compression methods, we integrated LoRC with both H2O and standard 8-bit quantization. Specifically, we allocated a 70% KV cache budget to LoRC and 85% to H2O, resulting in approximately a 60% overall cache budget. A similar configuration was used for the combination of LoRC with 8-bit quantization.

Table 12: Throughput and performance comparison of different methods..

| | Full Cache | H2O | KIVI | LoRC | LoRC w/ H2O | LoRC w/ 8-bit |
|---|---|---|---|---|---|---|
| Performance | 11.6 | 10.5 | 10.3 | 11.2 | 11.0 | 10.9 |
| s/it | 228.5 | 192.7 | 186.6 | 203.4 | 195.9 | 193.7 |

Our results show that LoRC achieves superior performance compared to existing compression methods while maintaining competitive throughput. While the SVD-based computations introduce some overhead, LoRC still has a higher throughput than its full cache baseline because we deployed a customized kernel to streamline the attention computation. It is important to note that our primary objective was not to develop a Pareto-optimal method (simultaneously optimizing both performance and throughput), but rather to introduce an orthogonal approach to KV cache compression and establish the progressive compression strategy.

Additionally, the integration of LoRC with both token-eviction and quantization methods demonstrates its compatibility with existing compression methods. With only a slight performance trade-off (11.0/10.9 vs 11.2), these hybrid configurations achieve better throughput compared to using LoRC alone, which shows the potential of combining LoRC with orthogonal compression approaches.

## K Enhancements in memory-constrained deployment scenarios

Table 13: LLaMA-3-8B-Instruct model on a single GPU with 24GB memory.

| Max Sequence Length | Batch Size | 8B w/ Full Cache | 8B w/ LoRC-60% |
|---|---|---|---|
| 2048 | 20 | OOM | 22.12 G |

Table 14: LLaMA-3-70B-Instruct model on 8 GPUs with 24GB memory.

| Max Sequence Length | Batch Size | 70B w/ Full Cache | 70B w/ LoRC-60% |
|---|---|---|---|
| 2048 | 8 | OOM | 22.97 G |

LoRC demonstrates significant benefits in memory-constrained deployment scenarios. To further validate this, we provide additional experiments here. These results demonstrate that LoRC enables the deployment of models that would otherwise be impossible to run with full KV cache on consumer GPUs.

