# OpenReview forum: "LoRC: Low-Rank Compression for LLMs KV Cache with a Progressive Compression Strategy"
_ICLR.cc/2025/Conference — Submitted to ICLR 2025_

### Official Review · Reviewer_CNeC · 2024-10-27

**Soundness:** 3
**Presentation:** 3
**Contribution:** 2
**Rating:** 6
**Confidence:** 4

**Summary:**

This paper introduces LoRC, a novel approach to KV cache compression that leverages the inherent low-rank properties of weight matrices. It employs a progressive layer-wise compression strategy to approximate the KV weight matrices without requiring model retraining. Experimental results demonstrate that LoRC significantly reduces GPU memory usage with minimal to no performance degradation.

**Strengths:**

1. The authors provide a theoretical analysis of error bounds for KV cache compression to examine a progressive compression strategy. The study indicates that compressing shallow layers less aggressively than deeper layers helps preserve model performance.

2. The authors propose a KV cache compression method using low-rank approximation of KV weight matrices for transformer-based (LLMs) without requiring model retraining.

3. The paper is well-organized and the motivation is clear.

**Weaknesses:**

1. By using low-rank approximation, the computational overhead should also be reduced compared to dense computation. In the experiments, I only found comparisons of compression ratio and performance. Could you provide details on your implementation for computing the attention layer? Did you use pure PyTorch? Additionally, have you tested the speedup compared to the full KV cache method?

2. The authors claims that the proposed methods enhances LLM performance in memory-constrained deployment scenarios. Are there any experiments to support it? For example, can this method be deployed on consumer GPUs with limited memory (24 GB or less) where the full KV cache model cannot fit? are there models that cannot be deployed on consumer GPUs but can be deployed using the proposed method?

**Questions:**

Please see the above.

---

> ### Author Response · Authors · 2024-11-25
> **Response to Reviewer CNeC**
>
> We appreciate the reviewers’ insightful feedback.
>
> > W1.1:  Details on implementation for computing the attention layer.
>
> **Response**:
> Within the attention layer, there are two major differences compared to the standard implementation. First, for the caching stage, we project keys to low-rank space without position embedding and store only the compressed representation. Second, for the attention computation, we load compressed keys and reconstruct them, apply RoPE, and then compute attention scores with position-aware keys.
>
> To deal with the compatibility with RoPE, the most straightforward implementation -- applying RoPE for each compressed key -- brings computational overhead. To address this, we follow H2O [1] and FastGen [2] to develop a customized kernel that fuses cache reconstruction and rotation operations. This approach minimizes memory transfers and computational overhead, enhancing the overall throughput.
>
> Specifically, we leverage Triton to implement a fused kernel that optimizes memory access and computation. With this kernel, we are able to combine key reconstruction from compressed representation and apply RoPE rotations on-the-fly. It also streamlines memory access because the single data load from global memory for compressed keys requires minimal data movement.
>
> With these engineering efforts, we achieve higher throughput compared to baseline configurations.
>
> > W1.2: Speedup compared to the full KV cache method.
>
> **Response**:
>
> Yes, we are able to achieve speedup compared to the full KV cache method. We present the throughput analysis below.
>
> Following H2O [1],  we conducted throughput experiments with fixed input and output sequence lengths using the LLaMA-3-70B-Instruct model on a node equipped with eight NVIDIA H100 80GB HBM3 GPUs. As discussed above, our method involves additional computations to recover the compressed cache and manage RoPE, and we developed a customized kernel that fuses cache reconstruction and rotation operations.
>
> | Input Length| Output Lenght | Batch Size | Full Cache Throughput  | LoRC-60% Throughput| Speedup |
> |:---------------:|:---------------:|:---------------:|:---------------:|:---------------:|:---------------:|
> |	1024	| 2048	|	32 	| 52.75 tokens / s	|	60.08 tokens / s | x 1.14 |
> |	1024	| 4096	|	32 	| 78.66 tokens / s | 	98.74 tokens / s | x 1.26 |
>
> The results above indicate that our engineered efforts can streamline the attention computation with LoRC compression, thereby achieving higher throughput compared to the full cache scenario.
>
> We also compare LoRC with other methods using the metric seconds per iteration (s/it) on XSum to obtain the real-world throughput. We adhered to the same 60% KV cache budget for both H2O and LoRC. In an effort to demonstrate compatibility with other KV cache compression methods, we integrated LoRC with both H2O and standard 8-bit quantization. Specifically, we allocated a 70% KV cache budget to LoRC and 85% to H2O, resulting in approximately a 60% overall cache budget. A similar configuration was used for the combination of LoRC with 8-bit quantization. For LoRC with quantization,  we also allocate 70% KV cache budget to LoRC and 85% to 8-bit quantization.
>
> | | Full Cache| H2O | KIVI | LoRC | LoRC w/ H2O | LoRC w/ 8-bit|
> |:---------------:|:---------------:|:---------------:|:---------------:|:---------------:|:---------------:|:---------------:|
> | Performance	| 11.6 | 10.5 | 10.3 | 11.2 | 11.0 | 10.9 |
> | s/it	| 228.5 | 192.7 | 186.6 | 203.4 | 195.9 | 193.7 |
>
> Our results show that LoRC achieves superior performance compared to existing compression methods while maintaining competitive throughput. While the SVD-based computations introduce some overhead, LoRC still has a higher throughput than its full cache baseline because we deployed a customized kernel to streamline the attention computation. It's important to note that our primary objective was not to develop a Pareto-optimal method (simultaneously optimizing both performance and throughput), but rather to introduce an orthogonal approach to KV cache compression and establish the progressive compression strategy.
>
> Additionally, the integration of LoRC with both token-eviction and quantization methods demonstrates its compatibility with existing compression methods. With only a slight performance trade-off (11.0/10.9 vs 11.2), these hybrid configurations achieve better throughput compared to using LoRC alone, which shows the potential of combining LoRC with orthogonal compression approaches.
>
> [1] H2O: Heavy-Hitter Oracle for Efficient Generative Inference of Large Language Models, NeurIPS 2023.
> [2] Model Tells You What to Discard: Adaptive KV Cache Compression for LLMs, ICLR 2024.

---

> ### Author Response · Authors · 2024-11-25
> **Response to Reviewer CNeC (2)**
>
> > W2: Enhancements in memory-constrained deployment scenarios.
>
> Yes, LoRC demonstrates significant benefits in memory-constrained deployment scenarios. To further validate this, we provide additional experiments here.
>
> - Deployment on Consumer GPUs (24GB Memory).
>
> | Max Sequence Length | Batch Size | 8B w/ Full Cache | 8B w/ LoRC-60% |
> |:---------------:|:---------------:|:---------------:|:---------------:|
> |2048 | 20 | OOM | 22.12 G |
>
> **Table R1**: LLaMA-3-8B-Instruct model on a single GPU with 24GB memory.
>
> | Max Sequence Length | Batch Size | 70B w/ Full Cache | 70B w/ LoRC-60% |
> |:---------------:|:---------------:|:---------------:|:---------------:|
> |2048 | 8 | OOM | 22.97 G |
>
> **Table R2**: LLaMA-3-70B-Instruct model on 8 GPUs with 24GB memory.
>
> These results demonstrate that LoRC enables the deployment of models that would otherwise be impossible to run with full KV cache on consumer GPUs.
>
> - Performance Preservation Under Aggressive Compression.
>
> In this set of experiments, we compare LoRC with token-eviction method (H2O) [1] and quantization method (KIVI) [3], using LLaMA-3-8B-Instruct. For H2O and LoRC, we keep the same 60% KV cache budget. For KIVI, we use the KIVI-4-bit implementation. We evaluate accuracy on BoolQ and OpenBook QA, and Rouge-Lsum for XSum.
> | Method | BoolQ | XSum  | OpenBook QA |
> |:---------------:|:---------------:|:---------------:|:---------------:|
> | Full Cache	| 81.6 | 11.6 | 78.0 |
> | H2O	| 76.4 | 10.5 | 75.1 |
> | KIVI 	| 77.6 | 10.3 | 74.8 |
> | LoRC  | 79.2 | 11.2 | 75.7 |
>
> These experiments demonstrate that LoRC not only enables deployment in memory-constrained environments but also maintains higher model performance compared to alternative approaches.
>
> - Additionally, as discussed in our first response to reviewer CNeC, LoRC improves system throughput while achieving these memory savings.
>
> [3] KIVI: A Tuning-Free Asymmetric 2bit Quantization for KV Cache, ICML 2024.

---

> > ### Comment · Reviewer_CNeC · 2024-11-26
> >
> > Thanks a lot for the valuable response and additional experiments. I will increase my rate.
> >
> > As you mentioned, your fused kernel is implemented with Triton, which enables TensorCore in GPU as the default setting. I am curious whether the compared methods also use TensorCore.

---

> > > ### Author Response · Authors · 2024-11-28
> > > **Response to Reviewer CNeC's Follow-up Comments**
> > >
> > > We deeply appreciate your engagement in the discussion period, and we are delighted that our response addressed your concerns and led to an increased rating.
> > >
> > > Regarding the TensorCore usage in compared methods, we illustrate the implementation details here:
> > > - For H2O, we use the implementation from cold-compress (github.com/AnswerDotAI/cold-compress), which leverages torch.compile for speedup through JIT compilation of PyTorch code.
> > > - For KiVi, we use their official implementation (github.com/jy-yuan/KIVI), which implements a group-wise quantization kernel in Triton.
> > > - The full cache baseline comes from the official Llama implementation using standard PyTorch. Compared to it, our Triton kernel specifically handles compressed key reconstruction, RoPE operations, and attention score calculation. The rest of operations in LoRC use standard PyTorch.
> > >
> > > We thank you for the follow-up again and please let us know if you have any additional questions.

---

### Official Review · Reviewer_P7NJ · 2024-10-28

**Soundness:** 4
**Presentation:** 3
**Contribution:** 3
**Rating:** 5
**Confidence:** 4

**Summary:**

This paper focuses on compressing the KV cache in long-context large language models. It cleverly utilizes Singular Value Decomposition (SVD) to decompose the KV cache and employs the Q and O matrices to absorb the decomposition, thereby reducing the required cache. The writing is clear, the proposed method is highly innovative, and experiments demonstrate the effectiveness of compressing the KV cache.

**Strengths:**

1. The core idea of the paper is insightful, as it proposes compressing the KV cache by performing low-rank decomposition on the KV values and absorbing them into the Q and O matrices.

2. Theoretical analysis and proofs are sufficient, as the paper derives upper and lower bounds for the error through mathematical formulations.

3. The paper demonstrates effective compression results, achieving high compression ratios of the KV cache with minimal impact on performance.

4. The writing is clear, making the paper easy to understand.

**Weaknesses:**

1. The author directly performs SVD decomposition and reconstruction on the K and V matrices. However, in previous SVD for LLM compression work, this original method of decomposing the weight matrix without re-training it usually leads to a large performance loss (as mentioned in Pufferfish, ASVD). The paper does not clearly explain why the performance loss is small under the same approach.

2. The paper does not prove the superiority of the proposed Progressive Compression Strategy. The increase in compression ratio with layer depth does not align with findings in existing work on SVD for LLM compression (e.g., ASVD, SVD-LLM, FWSVD).

3. The rationale for using the cumulative condition number as a compression metric for each layer is not clearly demonstrated.

4. The experiments are insufficient, as the paper lacks comparisons with previous KV cache compression methods and does not evaluate models other than Llama.

**Questions:**

1. In the previous work on SVD for LLM compression, directly performing SVD on the weight matrix without updating the weight matrix results in a huge performance loss. Can the author explain why there is no performance loss in this work?

2. Could the authors explain why using the Cumulative Condition Number as a criterion for rank selection is reasonable?

3. The progressive compression across layers seems questionable. Previous works on automatic rank pruning (e.g., ASVD, FWSVD) do not indicate that rank should decrease with increasing layer depth. Moreover, the final layers of a model are often crucial and should not have their rank excessively reduced. While the paper presents single-layer analysis, it does not consider inter-layer dependencies. Could the authors compare the Progressive Compression Strategy with other automatic rank pruning approaches to establish its validity or superiority?

4. Could the authors conduct experiments on more models (beyond Llama) and compare with other KV cache compression methods?

---

> ### Author Response · Authors · 2024-11-25
> **Response to Reviewer P7NJ**
>
> We appreciate your feedback and thoughtful review of our work.
>
>
> > Q1: Previous SVD works claim performance loss without updating the weight matrix, why the performance loss in this work is minimal?
>
> **Response**: Our work differs fundamentally from previous SVD works [1, 2, 3] in two key aspects:
>  - Scope of compression: [1, 2, 3] compress model parameters including more general weight matrices in LLMs, our KV cache compression approach specifically targets the Key and Value weight matrices in the attention module. Compressing only Kv matrices preserves the model's computational patterns in other components.
>  - Consideration of network-wide effects: previous works analyze each layer in isolation, while we consider the critical inter-layer dependencies. This explains why they report performance loss when the cumulative effects are ignored when compressing the matrices.
>
> Our key insight is that compression errors propagate differently through the network depending on where they originate. Our theoretical analysis (Section 5.3) rigorously proves that errors from earlier layers get amplified through network propagation, even if these layers individually show low sensitivity. That is why previous SVD approaches, which only consider single-layer properties, require weight updates or retraining to maintain performance. In contrast, our method's minimal performance loss stems from carefully accounting for these propagation effects through progressive compression and cumulative sensitivity analysis.
>
>
> [1] ASVD: Activation-aware Singular Value Decomposition for Compressing Large Language Models
> [2] Language model compression with weighted low-rank factorization
> [3] SVD-LLM: Truncation-aware Singular Value Decomposition for Large Language Model Compression
>
>
>
> > Q2: Why cumulative condition number is reasonable for rank selection?
>
> **Response**: The details of the derivation and illustration of cumulative condition number can be found in lines 188- 221. To summarize here, the cumulative condition number serves as a mathematically principled criterion for rank selection by capturing layer sensitivity from a network-wide perspective. Our approach is grounded in the following derivation:
>
> - Local Sensitivity: As shown in Eq. (11), a layer's acceptable input perturbation is inversely proportional to its condition number $\kappa(A_l)$. This provides a natural measure of a single layer sensitivity.
>
> - Error Propagation: In multi-layer transformers, compression-induced perturbations propagate through subsequent layers. By computing the product of condition numbers from the current layer through all subsequent layers (Eq. (12)), the cumulative condition number provides a holistic measure of each layer's impact on the final output.
>
> This formulation of the cumulative condition number is theoretically justified by our analysis in Section 5.3. It proves that errors in earlier layers pass through more subsequent transformations and nonlinearities, thus being amplified more significantly. The cumulative condition number directly quantifies this effect, making it an ideal metric for guiding compression decisions.
>
> > Q3: Progressive Strategy and Method Effectiveness.
>
> This question consists of multiple points, let us address each concern:
>
> 1. **Different Focus and Theory Compared to Prior Works**:
> While previous works (e.g., ASVD, FWSVD) compress entire model parameters, we specifically target KV weight matrices for cache compression. Unlike prior approaches that analyze layers in isolation, we consider the inter-layer dependencies from the network view. Our strategy is built on the intuition that compressed shallow layers could lead to cascading errors that propagate and amplify through the network, which is also supported by our theoretical analysis as presented in Section 5.3.
> Note that ASVD requires calibration data for perplexity calculation to determine layer sensitivity, and FWSVD requires post-compression finetuning. Our method only needs to access the model weights to determine layer sensitivity and work directly without post-compression model tuning -- this makes our approach more practical and widely applicable.
>
> 2. **Whether Compress Final Layer Aggressively**:
> The final layer’s importance could come from multiple components within that layer. Note that we only compress the key/value weight matrix of each layer. In Figure 4, we show that these matrices can be approximated with low reconstruction error, including the key/value weight matrix of the final layer. It means that compressing the final layer's key/value weight matrix via low-rank approximation is practical. Moreover, according to our derivation of the progressive compression strategy (Eq. 16), the error of this layer has the least propagation in the forward pass, thus it can be compressed most aggressively.

---

> ### Author Response · Authors · 2024-11-25
> **Response to Reviewer P7NJ (2)**
>
> > Q3: Progressive Strategy and Method Effectiveness.
>
> 3. **Single-layer and Inter-layer Analysis**:
> The inter-layer dependency means we consider the layer sensitivity from a network-wide perspective rather than analyze each layer alone, and we carefully consider the inter-layer dependencies throughout this paper. Theoretically, Section 5.3 provides a mathematical proof of error propagation patterns to illustrate how the inter-layer dependencies guide our design of the progressive compression strategy.
> Empirically, Section 6.6 especially validates the error propagation effects -- the noise caused by shallow compressed layers will be amplified more after propagation through the network.
>
> 4. **Superiority of the Progressive Compression Strategy**:
> Similar to the first point made in this question, previous methods need calibration data for truncation rank searching (ASVD) or post-compression finetuning (FWSVD), while our compression strategy works directly without any calibration data or post-compression model tuning. Additionally, the empirical results demonstrate that LoRC can preserve competitive performance compared to the full cache baseline with a significant compression ratio.
>
> > Q4.1:  Experiments on models beyond llama.
>
> We conduct more experiments using the OPT-30B and OPT-66B models on XSum and OpenBook QA (OBQA) to provide a more comprehensive evaluation of our method. We report Rouge-Lsum on XSum and accuracy on OBQA. The results are as shown below, where the percentage denotes the KV cache budget.
> | Model |  XSum - 100% | XSum - 80% | XSum - 60%   | OBQA - 100% | OBQA - 80% | OBQA - 60%  |
> |:---------------:|:---------------:|:---------------:|:---------------:|:---------------:|:---------------:|:---------------:|
> | OPT-30B	| 10.43 | 10.39 | 10.30 | 43.2 |  43.2 | 43.0 |
> | OPT-66B	| 10.70 | 10.72 | 10.65 | 44.3 | 43.6 | 43.4|
>
> The results demonstrate that:
> - Performance Preservation with Significant KV Cache Saving: Compared to the full-cache baseline, LoRC maintains model performance with negligible degradation even at significant KV cache compression levels.
> - Broad Applicability: Beyond the main results reported in our paper, these additional experiments demonstrate that LoRC is broadly applicable across diverse models, including LLaMA-2-13B, LLaMA-3-8B-Instruct, LLaMA-3-70B-Instruct, OPT-30B, and OPT-66B.
>
> > Q4.2: Comparison with Other KV Cache Compression Methods
>
> **Response**:
> We compare LoRC with the token-eviction method (H2O) and the quantization method (KIVI) using LLaMA-3-8B-Instruct. For H2O and LoRC, we keep the same 60% KV cache budget. For KIVI, we use the KIVI-4-bit implementation. We evaluate accuracy on BoolQ and OpenBook QA, and Rouge-Lsum for XSum. The results below show LoRC can preserve a better performance compared to token-eviction and quantization methods when KV cache is aggressively compressed.
> | Method | BoolQ | XSum  | OpenBook QA |
> |:---------------:|:---------------:|:---------------:|:---------------:|
> | Full Cache	| 81.6 | 11.6 | 78.0 |
> | H2O	| 76.4 | 10.5 | 75.1 |
> | KIVI 	| 77.6 | 10.3 | 74.8 |
> | LoRC  | 79.2 | 11.2 | 75.7 |
>
> For throughput comparison, we compare LoRC with other methods using the metric seconds per iteration (s/it) on XSum. We adhered to the same 60% KV cache budget for both H2O and LoRC. To demonstrate compatibility with other methods, we integrated LoRC with both H2O and standard 8-bit quantization. Specifically, we allocated a 70% KV cache budget to LoRC and 85% to H2O, resulting in approximately a 60% overall cache budget. A similar configuration was used for the combination of LoRC with 8-bit quantization. For LoRC with quantization,  we also allocate 70% KV cache budget to LoRC and 85% to 8-bit quantization.
>
> | | Full Cache| H2O | KIVI | LoRC | LoRC w/ H2O | LoRC w/ 8-bit|
> |:---------------:|:---------------:|:---------------:|:---------------:|:---------------:|:---------------:|:---------------:|
> | Performance	| 11.6 | 10.5 | 10.3 | 11.2 | 11.0 | 10.9 |
> | s/it	| 228.5 | 192.7 | 186.6 | 203.4 | 195.9 | 193.7 |
>
> Our results show that LoRC outperforms other methods while maintaining competitive throughput. While the SVD-based computations introduce some overhead, LoRC still has a higher throughput than its full cache baseline because we deployed a customized kernel to streamline the attention computation. Note that our primary objective was not to develop a Pareto-optimal method (simultaneously optimizing both performance and throughput), but rather to introduce an orthogonal approach to KV cache compression and establish the progressive compression strategy.
>
> Additionally, the integration of LoRC with both token-eviction and quantization methods demonstrates its compatibility with existing compression methods. With only a slight performance trade-off, these hybrid configurations achieve better throughput compared to using LoRC alone, which shows the potential of combining LoRC with orthogonal compression approaches.

---

> > ### Author Response · Authors · 2024-12-01
> > **Looking Forward to Reviewer P7NJ's Feedback**
> >
> > Dear Reviewer P7NJ,
> >
> > As we approach the end of the discussion period, we wanted to follow up regarding our previous responses to your valuable feedback.
> >
> > We have worked to thoroughly address your concerns via our responses as shown above. Besides, we have strengthened our paper with additional results, including latency and throughput analyses (https://openreview.net/forum?id=NI8AUSAc4i&noteId=21SkfMVh8A), as well as enhancements in memory-constrained scenarios (https://openreview.net/forum?id=NI8AUSAc4i&noteId=kQKX3GGgNx). We would greatly appreciate it if you could take a moment to review our responses and those additional results.
> >
> > With two days remaining in the discussion period, we welcome any additional questions or concerns you may have. If you find our responses and new results have addressed your initial concerns, we would appreciate your consideration in updating your assessment.
> >
> > Thank you again for your time and thoughtful review.

---

> > > ### Author Response · Authors · 2024-12-02
> > > **Kindly Awaiting Your Feedback**
> > >
> > > Dear Reviewer P7NJ,
> > >
> > > Thank you again for your valuable feedback. We would like to kindly remind you that **today (Dec 2nd, AoE)** is the last day that reviewers may post a message to the authors. We sincerely hope that our responses have enhanced the paper's quality and addressed your concerns, and we look forward to your feedback and potential updated rating.
> > >
> > > If there are any additional suggestions or comments you would like to provide, please don't hesitate to share them.
> > >
> > >
> > > Best Regards,
> > > Authors

---

### Official Review · Reviewer_9w6H · 2024-11-01

**Soundness:** 3
**Presentation:** 3
**Contribution:** 2
**Rating:** 3
**Confidence:** 4

**Summary:**

This proposes a method to reduce the memory demands of large language models (LLMs) by compressing KV caches using low-rank matrix decomposition on weights. LoRC designs a progressive compression strategy that adjusts compression rates across layers based on sensitivity to error propagation, thus providing better preservation of model performance.

This paper effectively designs a method for determining specific compression rates for each layer. However, it lacks critical measurements regarding practical acceleration and potential overheads in deployment. Without this information, it’s challenging to assess its suitability for real-world applications.

**Strengths:**

The progressive compression strategy effectively minimizes accuracy segregation with clear proof to demonstrate the error propagation under certain assumptions.

**Weaknesses:**

1. Limited Comparison with Related Work: The paper only compares LoRC to uniform compression, omitting relevant methods that leverage perplexity evaluation [1] or Hessian-based scoring [2] for deciding layer-wise compression rate for performing low-rank decomposition.
2. Lack of Measurements on Latency or Throughputs improvements: While LoRC demonstrates memory savings, it does not discuss potential gains in speed or latency—key metrics for assessing the real-world efficiency of compression methods.
3. Unclear Handling of Rotary Position Embedding (RoPE): The paper needs more clarity on the compatibility with RoPE. The author provides a brief section in Appendix C; the detailed workflow seems ambiguous, particularly on whether it must be reapplied at each step and what overhead this entails. Further explanation is crucial for models that rely on RoPE.

[1] ASVD: Activation-aware Singular Value Decomposition for Compressing Large Language Models
[2] Palu: Compressing KV-Cache with Low-Rank Projection

**Questions:**

1. How do you handle RoPE?
2. How much of this system is implemented and what speedups are you able to attain?

---

> ### Author Response · Authors · 2024-11-25
> **Response to Reviewer 9w6H**
>
> > W1: Comparison with Related Work.
>
> **Response**:
> While we appreciate the reviewer highlighting these relevant works, we respectfully disagree with this criticism regarding comparisons with [1] and [2].
> According to **ICLR 2025 guidelines** (https://iclr.cc/Conferences/2025/ReviewerGuide), authors are "not required to compare their own work to" very recent papers or non peer-reviewed (e.g., ArXiv) papers that are: a) Published after July 1, 2024 ;b) Not peer-reviewed; c) Only available on arXiv. Both [1] and [2] fall in the above categories. Actually, both of them are also under review of ICLR 2025.
> It would have been impossible for us to compare with these concurrent submissions.
>
> Nevertheless, we are open to a discussion of these approaches. Our method differs from [1] and [2] in two fundamental aspects:
> - Data-free Compression: ASVD [1] requires calibration data for perplexity calculation; PALU [2] needs a dataset to compute Fisher information. Our method requires only the model's weight matrices to determine layer sensitivity -- this makes our approach more practical and widely applicable.
>
> - Analysis of Inter-layer Dependencies: Both [1] and [2] analyze layer sensitivity in isolation. Our key innovation is considering inter-layer dependencies from a network-wide perspective. This is crucial because compression-induced perturbations in multi-layer transformers propagate through subsequent layers. Our progressive compression strategy provides a holistic measure of each layer's impact on the final output, and we theoretically prove its effectiveness through rigorous error propagation analysis (Section 5.3).
>
> These distinctions highlight our unique contributions.
>
> [1] ASVD: Activation-aware Singular Value Decomposition for Compressing Large Language Models, arxiv 2024.
> [2] Palu: Compressing KV-Cache with Low-Rank Projection, arxiv 2024.

---

> ### Author Response · Authors · 2024-11-25
> **Response to Reviewer 9w6H (2)**
>
> > W2 & Q2:  Latency & Throughputs Experiments.
>
> **Response**:
> Before addressing the latency and throughput results, we would like to emphasize that compressing KV cache itself is meaningful, because it enables the deployment of models that would otherwise be impossible to run with full KV cache on resource-constrained scenarios (e.g., consumer GPUs). In the main results of this paper, we aim to demonstrate that LoRC can preserve competitive performance compared to the full cache baseline when compressing aggressively.
>
> We present latency analysis and throughput experiments below.
>
> **Latency Analysis**:
>
> | Model | Computing Resource| Sensitivity Calculation (s) | SVD Processing (s) |
> |:---------------:|:---------------:|:---------------:|:---------------:|
> | LLaMA-3-8B-Instruct    |  NVIDIA H100 80GB HBM3 * 1 | 30.6  | 33.3   |
> | LLaMA-3-70B-Instruct  | NVIDIA H100 80GB HBM3 * 8 | 12.2   | 14.3   |
>
> The latency analysis is shown in the above table. Note that sensitivity calculation only takes once for a given model, and the SVD processing is also a one-time implementation during the model initialization stage. Such latency will not affect inference speed. Compared to the whole inference duration, the latency incurred by these processes is negligible. For example, the LLaMA-3-70B-Instruct model requires approximately 1 hour and 8 minutes to process 1,000 summaries from the XSUM dataset with a batch size of 32 and a sequence length of 8,000. The combined latency introduced by sensitivity calculations and SVD processing represents only **0.6%** of the total inference time.
>
> **Throughput Analysis**:
>
> Following H2O [3],  we conducted throughput experiments with fixed input and output sequence lengths using the LLaMA-3-70B-Instruct model on a node equipped with eight NVIDIA H100 80GB HBM3 GPUs. Because our method involves additional computations to recover the compressed cache and manage RoPE, we developed a customized CUDA kernel that fuses cache reconstruction and rotation operations. This approach minimizes memory transfers and computational overhead, enhancing the overall throughput.
> The results below indicate that our engineering efforts can streamline the attention computation with LoRC compression, thereby achieving higher throughput compared to full cache scenarios.
> | Input Length| Output Lenght | Batch Size | Full Cache Throughput  | LoRC-60% Throughput| Speedup |
> |:---------------:|:---------------:|:---------------:|:---------------:|:---------------:|:---------------:|
> |	1024	| 2048	|	32 	| 52.75 tokens / s	|	60.08 tokens / s | x 1.14 |
> |	1024	| 4096	|	32 	| 78.66 tokens / s | 	98.74 tokens / s | x 1.26 |
>
> We also compare LoRC with other methods [3, 4] using the metric seconds per iteration (s/it) on XSum to obtain the real-world throughput. For throughput analysis, we compared LoRC with other methods using the metric seconds per iteration (s/it) on XSum. We adhered to the same 60% KV cache budget for both H2O and LoRC. To demonstrate compatibility with other KV cache compression methods, we integrated LoRC with both H2O and standard 8-bit quantization. Specifically, we allocated a 70% KV cache budget to LoRC and 85% to H2O, resulting in approximately a 60% overall cache budget. A similar configuration was used for the combination of LoRC with 8-bit quantization. For LoRC with quantization,  we also allocate 70% KV cache budget to LoRC and 85% to 8-bit quantization.
>
> | | Full Cache| H2O | KIVI | LoRC | LoRC w/ H2O | LoRC w/ 8-bit|
> |:---------------:|:---------------:|:---------------:|:---------------:|:---------------:|:---------------:|:---------------:|
> | Performance	| 11.6 | 10.5 | 10.3 | 11.2 | 11.0 | 10.9 |
> | s/it	| 228.5 | 192.7 | 186.6 | 203.4 | 195.9 | 193.7 |
>
> Our results show that LoRC achieves superior performance compared to existing compression methods while maintaining competitive throughput. While the SVD-based computations introduce some overhead, LoRC still has a higher throughput than its full cache baseline because we deployed a customized kernel to streamline the attention computation. It's important to note that our primary objective was not to develop a Pareto-optimal method (simultaneously optimizing both performance and throughput), but rather to introduce an orthogonal approach to KV cache compression and establish the progressive compression strategy.
>
> Additionally, the integration of LoRC with both token-eviction and quantization methods demonstrates its compatibility with existing compression methods. With only a slight performance trade-off (11.0/10.9 vs 11.2), these hybrid configurations achieve better throughput compared to using LoRC alone, which shows the potential of combining LoRC with orthogonal compression approaches.
>
> [3] H2O: Heavy-Hitter Oracle for Efficient Generative Inference of Large Language Models, NeurIPS 2023.
> [4] KIVI: A Tuning-Free Asymmetric 2bit Quantization for KV Cache, ICML 2024.

---

> ### Author Response · Authors · 2024-11-26
> **Response to Reviewer 9w6H (3)**
>
> > W3 & Q1: How do you handle RoPE?
>
> **Response**:
> RoPE does bring additional computations to LoRC. To address this, we follow H2O [3] and FastGen [5] to develop a customized kernel that fuses cache reconstruction and rotation operations. This approach minimizes memory transfers and computational overhead, enhancing the overall throughput.
>
> Specifically, we leverage Triton to implement a fused kernel that optimizes memory access and computation. With this kernel, we are able to combine key reconstruction from compressed representation and apply RoPE rotations on-the-fly. It also streamlines memory access because the single data load from global memory for compressed keys requires minimal data movement.
>
> In a complete RoPE workflow, there are two stages
> - Caching: We project keys to low-rank space without position embedding and store only the compressed representation.
> - Attention computation: Within the customized kernel, we load compressed keys into shared memory, reconstruct and apply RoPE in a fused operation, and then compute attention scores with position-aware keys.
>
> With these engineering efforts, we achieve higher throughput compared to baseline configurations.
>
> [5] Model Tells You What to Discard: Adaptive KV Cache Compression for LLMs, ICLR 2024.

---

> > ### Comment · Reviewer_9w6H · 2024-11-27
> > **Thank you for a detailed response but I still have concerns**
> >
> > - Parallel work: Regarding parallel/arxiv work. I withdraw my concern there. While the cited work is highly relevant, it is indeed not published yet. In fact, I found that both works were submitted to this same conference. No need to compare against them although both ASVD and Palu are relevant points of comparison to your work moving forward.
> >
> > - GPU implementation: I firmly believe that the practical implications of implementing SVN on KV-cache are important. While your rebuttal mentions a fused Triton kernel implementation that takes RoPe embeddings into account, your manuscript does not mention this at all. It is tricky to get this kernel to perform well with high performance, especially with ROPE embeddings that need to be applied to reconstructed keys (not low-rank). Your rough description of this in the rebuttal makes sense to me but this is a system-level contribution that is simply missing from the paper.
> >
> > I would also like to point out that you mention *custom CUDA kernel* in your response (2) but *custom Triton kernel* in response (3) which reduces my confidence in your rebuttal.

---

> ### Author Response · Authors · 2024-11-28
> **Response to Reviewer 9w6H's Follow-up Comments**
>
> Thank you so much for your engagement in this discussion period. We organize our responses below.
>
> >C1: Comparison with concurrent work.
>
> We thank reviewer 9w6H for withdrawing their concern about concurrent work comparisons. Rather than evading discussion, we welcome the opportunity to include these relevant concurrent works, as they help highlight two key advantages of our approach:
> - First, LoRC achieves data-free compression without any calibration data or profiling data for determining compression extents [1, 2]. Instead, LoRC operates purely on weight matrices, which makes it immediately deployable, especially in scenarios where additional data access is restricted or computationally expensive.
> - Second, different from the related works that analyze layer sensitivity in isolation, we consider inter-layer dependencies from a network-wide perspective and propose the progressive compression strategy accordingly. We also provide a theoretical analysis of error propagation through transformer layers that rigorously support this progressive design, as acknowledged by reviewer **vHmh**, **P7NJ**, and **CNeC** in their mentioned strengths.
>
> >C2: GPU Implementation, Throughput Performance, and Writing.
>
> This comment consists of several points, we organize our response in the following aspects.
>
> **Clarification Request**: Before addressing specific points, we would like to seek clarification on what is meant by "SVN" - we assume this may refer to SVD, which is the technique used in our work.
>
> **Paper Positioning and Core Contributions**: We want to emphasize that we are not positioning our paper as a throughput-SOTA compression method, as SVD-based compression inherently introduces computational overhead. Instead, our key contribution lies in the design of an orthogonal compression technique, the progressive compression strategy, the corresponding theoretical foundations, and its ability to preserve performance under aggressive compression.
>
> **Throughput Performance**: As shown in our throughput analysis made in the first-round response (https://openreview.net/forum?id=NI8AUSAc4i&noteId=21SkfMVh8A), while our Triton-based implementation helps LoRC achieve 1.14-1.26× speedup over the full cache baseline, it still faces inherent computational overhead compared to throughput-optimized methods. However, our experiments reveal LoRC's strong compatibility with both token-eviction and quantization methods. To pursue throughput enhancement, the integration with H2O or 8-bit quantization achieves better throughput (195.9 s/it and 193.7 s/it vs 203.4 s/it) with minimal performance impact (11.0/10.9 vs 11.2). Regarding the comment "It is tricky to get this kernel to perform well with high performance," we believe future work can either explore compression methods integration or more optimized kernel-level implementation to further advance the throughput.
>
> **Terminology Clarification and Writring Focus**: Regarding terminology, we acknowledge the confusion in using CUDA kernel and Triton kernel interchangeably. To clarify, our implementation uses Triton for kernel development. It is great to know such implementation draws interest from this reviewer, but our original manuscript focused on LoRC's core algorithmic and theoretical contributions. Following the reviewer's suggestion, we are revising the paper to include these engineering efforts and to demonstrate LoRC's compatibility with existing throughput-optimized methods.
>
>
> We again sincerely appreciate the reviewer's engagement in the discussion stage. Please let us know if there are any additional concerns or questions.

---

### Official Review · Reviewer_vHmh · 2024-11-03

**Soundness:** 3
**Presentation:** 3
**Contribution:** 3
**Rating:** 6
**Confidence:** 3

**Summary:**

The paper introduces a low-rank compression approach (LORC) for large language models (LLMs) to efficiently manage the Key-Value (KV) cache, crucial for faster inference but memory-intensive. Traditional KV compression strategies either require model retraining, limiting practical deployment. LORC tackles this with a low-rank approximation of KV matrices, integrating progressively across layers to minimize error propagation. Experiments demonstrate substantial memory savings with minimal performance loss, providing a scalable, plug-and-play solution adaptable to pre-trained models without needing task-specific tuning.

**Strengths:**

1. The process of calculating sensitivity and condition number for each layer in LoRC is novel.

2. Mathematically robust proofs support the proposed method.

3. Compared to a uniform compression rate, the proposed progressive compression strategy achieves higher accuracy at the same compression rate.

**Weaknesses:**

**1. Latency and Throughput Experiments**

   The primary purpose of reducing KV Cache size is to decrease memory usage during inference, which ideally should lower latency or increase throughput. However, the LoRC method may introduce additional computational overhead due to:

- The need to recover the compressed KV cache, which requires extra computations.
- Frequent reading and writing of the large restored KV cache per layer for attention calculations, potentially impacting latency or throughput.
- The time required to compute sensitivity and perform SVD.

Please provide experimental results showing LoRC’s latency and throughput, including the time required for sensitivity calculations and SVD processing.

**2. Comparison with Other KV Cache Compression Methods**
   - The paper would be strengthened by a comparative analysis with other existing KV cache compression methods, including:
      - KV Cache eviction [1]
      - KV Cache quantization [2]
      - SVD-LLM [3]

Please compare LoRC's accuracy and latency at the same compression rates against these methods to highlight its unique benefits and trade-offs.

**3. (minor) Different trends in sensitivity by compression between training and inference.**

 Section 6.6 analyzes that compression in earlier layers negatively impacts inference due to the amplification of input-induced errors through forward propagation. In contrast, it is known that during training, compression in later layers tends to negatively impact accuracy [4-6]. Although the results are opposite, I think the author’s argument is reasonable as input errors amplify forward during forward propagation while input gradient errors amplify backward during backpropagation. Reflecting this point in Section 6.6 or the appendix could further strengthen the argument of the paper.

[1] H2O: Heavy-Hitter Oracle for Efficient Generative Inference of Large Language Models, Neurips, 2023.
[2] ZipCache: Accurate and Efficient KV Cache Quantization with Salient Token Identification, arxiv, 2024.
[3] SVD-LLM: Truncation-aware Singular Value Decomposition for Large Language Model Compression, arixv, 2024.
[4] GACT: Activation Compressed Training for Generic Network Architectures, ICML, 2022.
[5] ALAM: Averaged Low-Precision Activation for Memory-Efficient Training of Transformer Models, ICLR, 2024.
[6] DropBP: Accelerating Fine-Tuning of Large Language Models by Dropping Backward Propagation, Neurips, 2024.

**Questions:**

The primary concern is related to the weaknesses above. If these concerns are adequately answered, I am willing to consider increasing the score to 6.

---

> ### Author Response · Authors · 2024-11-25
> **Response to Reviewer vHmh**
>
> We appreciate the reviewer’s insightful comments, we organize our responses as below.
>
>
> > Q1: Latency and Throughput Experiments
>
>  **Response**:
>
> **Latency Analysis**:
>
> | Model | Computing Resource| Sensitivity Calculation (s) | SVD Processing (s) |
> |:---------------:|:---------------:|:---------------:|:---------------:|
> | LLaMA-3-8B-Instruct    |  NVIDIA H100 80GB HBM3 * 1 | 30.6    | 33.3    |
> | LLaMA-3-70B-Instruct  | NVIDIA H100 80GB HBM3 * 8 | 12.2    | 14.3    |
>
> The latency analysis is shown in the above table. Note that sensitivity calculation only takes once for a given model, and the SVD processing is also a one-time implementation during the model initialization stage. Such latency will not affect inference speed. Compared to the whole inference duration, the latency incurred by these processes is negligible. For example, the LLaMA-3-70B-Instruct model requires approximately 1 hour and 8 minutes to process 1,000 summaries from the XSUM dataset with a batch size of 32 and a sequence length of 8,000. The combined latency introduced by sensitivity calculations and SVD processing represents only 0.6% of the total inference time.
>
> **Throughput Analysis**:
> Our method involves additional computations to recover the compressed cache and manage RoPE. To address this, we developed a customized kernel that fuses cache reconstruction and rotation operations. This approach minimizes memory transfers and computational overhead, enhancing the overall throughput.
>
> Following H2O [1],  we conducted throughput experiments with fixed input and output sequence lengths using the LLaMA-3-70B-Instruct model on a node equipped with eight NVIDIA H100 80GB HBM3 GPUs. The results below indicate that our engineering efforts can streamline the attention computation with LoRC compression, thereby achieving higher throughput compared to full cache scenarios.
> | Input Length| Output Lenght | Batch Size | Full Cache Throughput  | LoRC-60% Throughput| Speedup |
> |:---------------:|:---------------:|:---------------:|:---------------:|:---------------:|:---------------:|
> |	1024	| 2048	|	32 	| 52.75 tokens / s	|	60.08 tokens / s | x 1.14 |
> |	1024	| 4096	|	32 	| 78.66 tokens / s | 	98.74 tokens / s | x 1.26 |
>
>
> [1] H2O: Heavy-Hitter Oracle for Efficient Generative Inference of Large Language Models, NeurIPS 2023.

---

> ### Author Response · Authors · 2024-11-25
> **Response to Reviewer vHmh (2)**
>
> > Q2: Comparison with Other KV Cache Compression Methods
>
> **Response**:
> We aimed to include all three suggested works in our comparison. However, we successfully incorporated only H2O due to limitations with the other methods. ZipCache, which has not yet released the full code necessary for running inference, was replaced by KIVI [2], another method that employs KV cache quantization. SVD-LLM, which includes a calibration stage using additional data to update the compressed matrix, was deemed non-comparable to our experimental setting due to this difference.
>
> We conduct experiments using LLaMA-3-8B-Instruct. For H2O and LoRC, we keep the same 60% KV cache budget. For KIVI, we use the KIVI-4-bit implementation. We evaluate accuracy on BoolQ and OpenBook QA, and Rouge-Lsum for XSum. The results below show LoRC can preserve a better performance compared to token-eviction and quantization methods when KV cache is aggressively compressed.
> | Method | BoolQ | XSum  | OpenBook QA |
> |:---------------:|:---------------:|:---------------:|:---------------:|
> | Full Cache	| 81.6 | 11.6 | 78.0 |
> | H2O	| 76.4 | 10.5 | 75.1 |
> | KIVI 	| 77.6 | 10.3 | 74.8 |
> | LoRC  | 79.2 | 11.2 | 75.7 |
>
> For throughput analysis, we compared LoRC with other methods using the metric seconds per iteration (s/it) on XSum. We adhered to the same 60% KV cache budget for both H2O and LoRC. In an effort to demonstrate compatibility with other KV cache compression methods, we integrated LoRC with both H2O and standard 8-bit quantization. Specifically, we allocated a 70% KV cache budget to LoRC and 85% to H2O, resulting in approximately a 60% overall cache budget. A similar configuration was used for the combination of LoRC with 8-bit quantization. For LoRC with quantization,  we also allocate 70% KV cache budget to LoRC and 85% to 8-bit quantization.
>
> | | Full Cache| H2O | KIVI | LoRC | LoRC w/ H2O | LoRC w/ 8-bit|
> |:---------------:|:---------------:|:---------------:|:---------------:|:---------------:|:---------------:|:---------------:|
> | Performance	| 11.6 | 10.5 | 10.3 | 11.2 | 11.0 | 10.9 |
> | s/it	| 228.5 | 192.7 | 186.6 | 203.4 | 195.9 | 193.7 |
>
> Our results show that LoRC achieves superior performance compared to existing compression methods while maintaining competitive throughput. While the SVD-based computations introduce some overhead, LoRC still has a higher throughput than its full cache baseline because we deployed a customized kernel to streamline the attention computation. It's important to note that our primary objective was not to develop a Pareto-optimal method (simultaneously optimizing both performance and throughput), but rather to introduce an orthogonal approach to KV cache compression and establish the progressive compression strategy.
>
> Additionally, the integration of LoRC with both token-eviction and quantization methods demonstrates its compatibility with existing compression methods. With only a slight performance trade-off (11.0/10.9 vs 11.2), these hybrid configurations achieve better throughput compared to using LoRC alone, which shows the potential of combining LoRC with orthogonal compression approaches.
>
> [2] KIVI: A Tuning-Free Asymmetric 2bit Quantization for KV Cache, ICML 2024.
>
> > Q3: Different trends in sensitivity by compression between training and inference.
>
> We appreciate the reviewer's insightful observation about the contrasting sensitivity trends between training and inference. This is because forward propagation amplifies early-layer compression errors during inference, while backward propagation amplifies later-layer compression effects during training.
>
> To provide a detailed discussion, we can compare the error propagation mechanisms.
> In our method (inference-stage compression), errors from compressed earlier layers propagate forward through the network. As shown in our Theorem 3 (Section 5.3), the error at network output is bounded by Eq.16, where earlier layer errors get amplified more because they pass through more subsequent transformations and nonlinearities.
>
> For those training-stage compression methods as introduced in [4-6], their rationale is that gradient errors propagate backward through the network. For example, DropBP [6] measure the layer sensitivity by:
> $S_l = \sum_i ( \|\nabla \mathbf{W}_i \|_2 - \|\nabla \mathbf{W}_i^{(l)} \|_2 )^2$
> where $S_l$ denotes the sensitivity of the l-th layer. Here, $\( \nabla \mathbf{W}_i \)$ represents the parameter gradient of the \( i \)-th layer when no layers are dropped, while $\( \nabla \mathbf{W}_i^{(l)} \)$ denotes the parameter gradient of the $i$-th layer when the $l$-th layer is dropped during backward propagation.
> It shows that later layers have higher impact on gradients propagating backward to all previous layers.
>
> We will add this discussion to the appendix and add the suggested works in related works.

---

> > ### Comment · Reviewer_vHmh · 2024-11-28
> > **Response to Authors**
> >
> > Thank you for the detailed experiments and responses. The experiments on throughput were particularly impressive. After carefully reviewing all the authors' responses, I have decided to raise my score to 6. However, I hope the final version of the paper reflects the detailed explanations provided, including the overhead from SVD, throughput results, and trends in sensitivity analysis, to make the paper even more remarkable.

---

> > > ### Author Response · Authors · 2024-11-28
> > > **Response to Reviewer vHmh's Follow-up Comments**
> > >
> > > We sincerely thank you for your engagement during the discussion period and your positive feedback on our responses. We are thrilled to know our responses addressed your concerns and contributed to an increased rating.
> > >
> > > In the revised version of the paper, we have included the new results in the appendix and plan to further refine the discussions on related works in the final version to highlight the trends in sensitivity analysis as you mentioned.
> > >
> > > Once again, we appreciate your constructive input and support in making our paper stronger.

---

### Author Response · Authors · 2024-12-02
**Global Response and Kind Reminder**

We are grateful to all the reviewers for their time and efforts in the review process. To provide a clear overview of our clarifications, supplementary results, and additional details, we have summarized the major updates made in our responses and the revised manuscript.

1. Latency analysis (**Appendix D**);
2. Throughput analysis (**Appendix E**);
3. Comparison with other KV cache methods (**Table 6, Appendix F**);
4. Compatibility with other KV cache methods (**Table 7, Appendix F**)
5. Enhancements in memory-constrained deployment scenarios (**Appendix G**)
6. Engineering details - RoPE handling for position embedding compatibility and customized kernel implementation for computational efficiency (**Appendix C**);
7. Different sensitivity trends between training-stage compression and inference-stage compression (**Response to Reviewer vHmh (2) - Q3**, https://openreview.net/forum?id=NI8AUSAc4i&noteId=l5ocEi1SlZ);
8. Progressive compression strategy’s rationale and effectiveness (**Response to Reviewer P7NJ**, https://openreview.net/forum?id=NI8AUSAc4i&noteId=UoGP0BNNu2).

Thank you once again for your engagement during the review process. For those who haven’t had the opportunity to review our responses, we understand it takes time to review multiple submissions, and we are still looking forward to your feedback to reflect the discussion period.

---

### Meta-Review · Area_Chair_vxXf · 2024-12-20

**Metareview:**

The paper introduces LoRC (Low-Rank Compression), a method for compressing the Key-Value (KV) cache in large language models (LLMs) using low-rank matrix decomposition of the KV projection matrices. Key contributions include:

- A progressive compression strategy that varies compression rates across layers based on layer sensitivity to error propagation.

- Theoretical analysis of error bounds to justify the compression strategy and its impact on preserving model performance.

- Demonstration of practical applicability: LoRC is compatible with existing transformer-based LLMs and does not require retraining.

The reviewers praised the novelty of the idea and particularly the associated theoretical analysis of error bounds that guided the design of the progressive compression strategy. In addition, the method seamlessly integrates with existing LLMs without retraining which makes it practical for real-world deployment. The paper is also well organized and presented.

A major concern comes from reviewer 9w6H who raised questions on the lack of a kernel implementation for handling ROPE embeddings, which has important practical implications. The authors acknowledged this limitation but argued that their contribution lies in the modeling change and the theoretical justifications. However, upon reviewing the paper I found that the modeling contributions may have important shortcomings.

- The paper overlooked the closely related works ASVD, FWSVD, SVD-LLM (as raised by the reviewers), and LoSparse (see below). Many of these works provided improved versions of SVD based model compression, and their absence diminishes the contextual grounding of the proposed approach.

- Sensitivity analysis underpinning the key theoretical results of the paper lacks discussion of related studies, particularly those in network robustness [2, 3]. In particular, the analysis of the paper only provides the naive upper bound but more precise calculations of the Lipschitze constant [4, 5] exist in the literature. Such existing results may also benefit the method presented in this paper.

- There is no clear justification for the cumulative condition number in Eq. (12), which is the key factor in the proposed algorithm. While this definition only considers key and value matrices, the propagation of error is affected by all the other layers as well hence should be taken into account?

Given these shortcomings, I recommend a weak rejection of the current version. Meanwhile, I strongly recommend the authors to revise their manuscript and further develop their method based on reviewer comments. I believe the paper has significant potential to make a valuable contribution to the field.

[1] LoSparse: Structured Compression of Large Language Models based on Low-Rank and Sparse Approximation

[2] Estimating Neural Network Robustness via Lipschitz Constant and Architecture Sensitivity

[3] Formalizing Generalization and Adversarial Robustness of Neural Networks to Weight Perturbations

[4] Lipschitz regularity of deep neural networks: analysis and efficient estimation

[5] LIPSCHITZ CONSTANT ESTIMATION OF NEURAL NETWORKS VIA SPARSE POLYNOMIAL OPTIMIZATION

**Additional Comments On Reviewer Discussion:**

In the discussion phase, the reviewers raised valuable questions on various aspects of the experiment.  The authors provided a significant amount of experiments on e.g. testing on additional models, latency analysis, comparison with related methods, which addressed the reviewer concerns.

---

### Decision · Program_Chairs · 2025-01-22

Reject